# Single-cell transcriptional profile of CD34+ hematopoietic progenitor cells from del(5q) myelodysplastic syndromes and impact of lenalidomide

While myelodysplastic syndromes with del(5q) (del(5q) MDS) comprises a well-defined hematological subgroup, the molecular basis underlying its origin remains unknown. Using single cell RNA-seq (scRNA-seq) on CD34+ progenitors from del(5q) MDS patients, we have identified cells harboring the deletion, characterizing the transcriptional impact of this genetic insult on disease pathogenesis and treatment response. Interestingly, both del(5q) and non-del(5q) cells present similar transcriptional lesions, indicating that all cells, and not only those harboring the deletion, may contribute to aberrant hematopoietic differentiation. However, gene regulatory network (GRN) analyses reveal a group of regulons showing aberrant activity that could trigger altered hematopoiesis exclusively in del(5q) cells, pointing to a more prominent role of these cells in disease phenotype. In del(5q) MDS patients achieving hematological response upon lenalidomide treatment, the drug reverts several transcriptional alterations in both del(5q) and non-del(5q) cells, but other lesions remain, which may be responsible for potential future relapses. Moreover, lack of hematological response is associated with the inability of lenalidomide to reverse transcriptional alterations. Collectively, this study reveals transcriptional alterations that could contribute to the pathogenesis and treatment response of del(5q) MDS.

Deletion of the long arm of chromosome 5 (del(5q)) is the most frequently observed cytogenetic alteration in de novo myelodysplastic syndromes (MDS). It affects around 10-15% of MDS patients and represents a distinct hematological and pathological subgroup due to its unique clinical features, such as macrocytosis, anemia, normal or high platelet count, and hypolobulated megakaryocytes[1]. With the advent of high-throughput sequencing technologies, two commonly deleted regions (CDR) were identified and mapped to del(5q): the 1.5 megabase region at 5q32-q33, which contains 41 coding genes[2,3], and the 5q31 region containing 45 coding genes. Thus, the characterization of the molecular pathogenesis of del(5q) MDS has mainly focused on

the identification of genes within these CDRs that are involved in the pathophysiology of the disease, such as *RPS14*, *miR-145*, *miR-146* and *EGR1*[4–7]. However, the presence of gene regulatory networks (GRNs) altered in these patients, as well as the transcriptional profile of hematopoietic progenitors from patients with del(5q), have not been analyzed in detail, in part due to the difficulty of identifying cells with and without the deletion within the same patient.

Lenalidomide represents the first therapeutic approach for del(5q) MDS, and induces in these patients a prolonged red blood cell transfusion independence and cytogenetic response[8]. Interestingly, even at the time of complete clinical and cytogenetic remission,

✉e-mail: fprosper@unav.es; tezponda@unav.es; mhernaez@unav.es

persistent del(5q) progenitor cells have been identified, providing a reasonable explanation for the loss of responsiveness that patients experience in a 2-to 3-year interval[9]. However, the real impact of the deletion during lenalidomide treatment remains unknown, hinting the need of characterizing the molecular mechanisms driven by the deletion that could be impairing treatment response.

Previous studies have demonstrated that transcriptional alterations play a key role in MDS pathogenesis[10–14]. For example, our group recently identified the transcription factor *DDIT3* as a key erythropoietic regulator that is overexpressed in MDS, and showed its potential as a therapeutic target[15]. Furthermore, gene expression profiles have shown to be affected by different types of alterations, including cytogenetic abnormalities and mutations, among other factors[16,17]. In this sense, several studies have performed gene expression analyses in del(5q) MDS, and have identified several pathways deregulated in the disease, including those related to Wnt/β catenin or integrin signaling, as well as genes that could be potentially contributing to disease pathogenesis, such as *SPARC*[14,18]. Other studies have depicted the effect of lenalidomide on the gene expression profile of del(5q) patients, evidencing its potential to restore erythroid differentiation, modulate the bone marrow microenvironment, and revert the aberrant expression of putative pathogenic microRNAs[19–22]. However, the hematopoietic system of these patients is composed of a mixture of cells with and without the deletion (termed del(5q) and non-del(5q) cells, respectively), which masks the expression profile of the del(5q) cells, and limits the ability to define the transcriptional impact of the deletion.

In this work, we performed single-cell RNA-sequencing (scRNA-seq) in primary CD34+ hematopoietic progenitor cells from del(5q) MDS patients at diagnosis and patients after lenalidomide treatment, and applied copy number alteration (CNA) analyses to link the del(5q) genotype to the transcriptional profile of each individual cell. This approach yielded a well-characterized del(5q) MDS atlas. Leveraging the generated atlas, we detected transcriptional alterations both in del(5q) cells and also in non-del(5q) cells at diagnosis and after lenalidomide treatment. We demonstrate that non-del(5q) cells present aberrant behavior compared to the healthy hematopoietic system similar to that observed in del(5q) cells. We also show that although lenalidomide restores some of the detected alterations of del(5q) and non-del(5q) cells of responder patients, other lesions identified at diagnosis remain after treatment. Furthermore, our results evidence that lenalidomide is not able to reverse part of the transcriptional lesions carried by del(5q) cells of a non-responder patient, which seems to be associated with the lack of hematological response.

## Results

### Single-cell RNA-sequencing of hematopoietic progenitor cells of del(5q) MDS patients
To identify the transcriptional alterations characterizing hematopoietic progenitors harboring del(5q), we initially performed scRNA-seq of CD34+ cells of four newly diagnosed patients with del(5q) MDS (Patient_1-4), and three age-matched healthy donors (Healthy_1-3) using the 10X Genomics technology (Fig. 1A) (gating strategy can be found in Supplementary Fig. 1). The clinical and genomic characteristics of the MDS patients and healthy donors are shown in Supplementary Table 1. The percentage of cells with del(5q) based on the cytogenetic analysis varied between 35 to 90%. In all cases, the common deleted region encompassed bands 5q(13-33) (genes shown in Supplementary Data 1).

A total of 55,119 and 45,311 cells from patients and healthy donors, respectively, were profiled and integrated. After applying quality filters, 46,772 and 43,442 cells were eventually included in the downstream analysis. Data was integrated, clustered and manually annotated (Fig. 1B, C, Supplementary Fig. 2A, B) based on curated markers (Fig. 1D, Supplementary Fig. 2C), obtaining 14 and 13 clusters (patients and donors, respectively) representing all the expected hematopoietic progenitor subtypes. Contribution of every MDS patient and donor to the composition of all the clusters was identified (Fig. 1E, Supplementary Fig. 2D), and each individual showed different proportions of hematopoietic progenitors (Fig. 1F, Supplementary Fig. 2E). Although there were some differences in the percentage of hematopoietic progenitors between MDS patients and healthy donors (e.g., HSC), these differences were not statistically significant which might be related to the high variability in cell composition across samples (Fig. 1G).

### Identification of CD34+ cells harboring del(5q) in MDS patients
Identifying single-arm copy number variations (CNVs) at the single-cell level presents challenges due to potential compensatory mechanisms of alleles, as well as to the sparse and noisy nature of single-cell data. In this study, we employed two different and complementary approaches: CopyKat[23] (Fig. 2A), which relies on gene expression, and CaSpER[24] (Fig. 2B), which relies on allele frequencies (see Methods). This combined strategy aimed to enhance the sensitivity and accuracy of identifying cells harboring 5q deletion. To avoid false positive detection, we only classified the cells as harboring the del(5q) if the same cell was characterized as such by the two different algorithms (Fig. 2C). To validate this classification, we analyzed the expression pattern of genes encoded in the deleted region in individual cells. Due to the sparsity of scRNA-seq data, we were only able to detect six genes as highly variable, *CD74*, *RPS14*, *BTF3*, *COX7C*, *HINT1* and *RPS23*, whose expression was decreased in del(5q) when compared to non-del(5q) cells at sample level (Fig. 2D), further confirming our del(5q) cell classification. Once the classification was performed, we applied a Wilcoxon signed-rank test between cells classified as del(5q) and non-del(5q), revealing in the underexpressed fraction of the genes an enrichment for the genes located on the deleted locus (Supplementary Fig. 3A, B). To further validate the classification, we randomly shuffled the labels from the classified cells, and repeated the same differential expression analysis, revealing how the genes located on the deleted region started to fade away (Supplementary Fig. 3C). Based on this classification, interestingly, for each individual patient, the proportion of del(5q) in the CD34+ progenitor cells was consistent with that obtained by karyotype in total bone marrow (Fig. 2E).

We then interrogated the distribution of del(5q) cells across the different hematopoietic progenitors. Cells with the deletion were detected in all the defined hematopoietic progenitor clusters (Fig. 3A), although a high heterogeneity of distribution was observed among patients (Fig. 3B, C). Despite the observed heterogeneity, a statistically significant accumulation of del(5q) cells was detected in early erythroid progenitors across all individuals (hypergeometric test, *p*-value < 0.05). Additionally, three out of four patients exhibited statistically significant enrichment of del(5q) in granulocyte-monocyte progenitors (GMP), megakaryocyte and late erythroid progenitors (Fig. 3D). Collectively, these results indicated a bias of del(5q) cells towards specific myeloid compartments, mainly towards erythroid cells, which is consistent with the association between this genetic lesion and the anemia that characterizes patients with del(5q) MDS.

### Transcriptional differences between del(5q) and non-del(5q) cells within MDS patients are driven by few specific transcriptional programs playing key roles in MDS
To delve into the transcriptional program associated with del(5q) cells in patients with MDS, we performed a pseudobulk differential expression (DE) analysis between del(5q) and non-del(5q) cells for each cell population, as traditional Wilcoxon signed-rank test -based DE analysis in single cell data has recently shown to yield high false positive rates[25]. Intriguingly, considering every type of hematopoietic progenitors, only seven genes were differentially expressed (downregulated) in del(5q) in comparison with non-del(5q) cells (Fig. 4A). Some of the downregulated genes played a key role in MDS and other

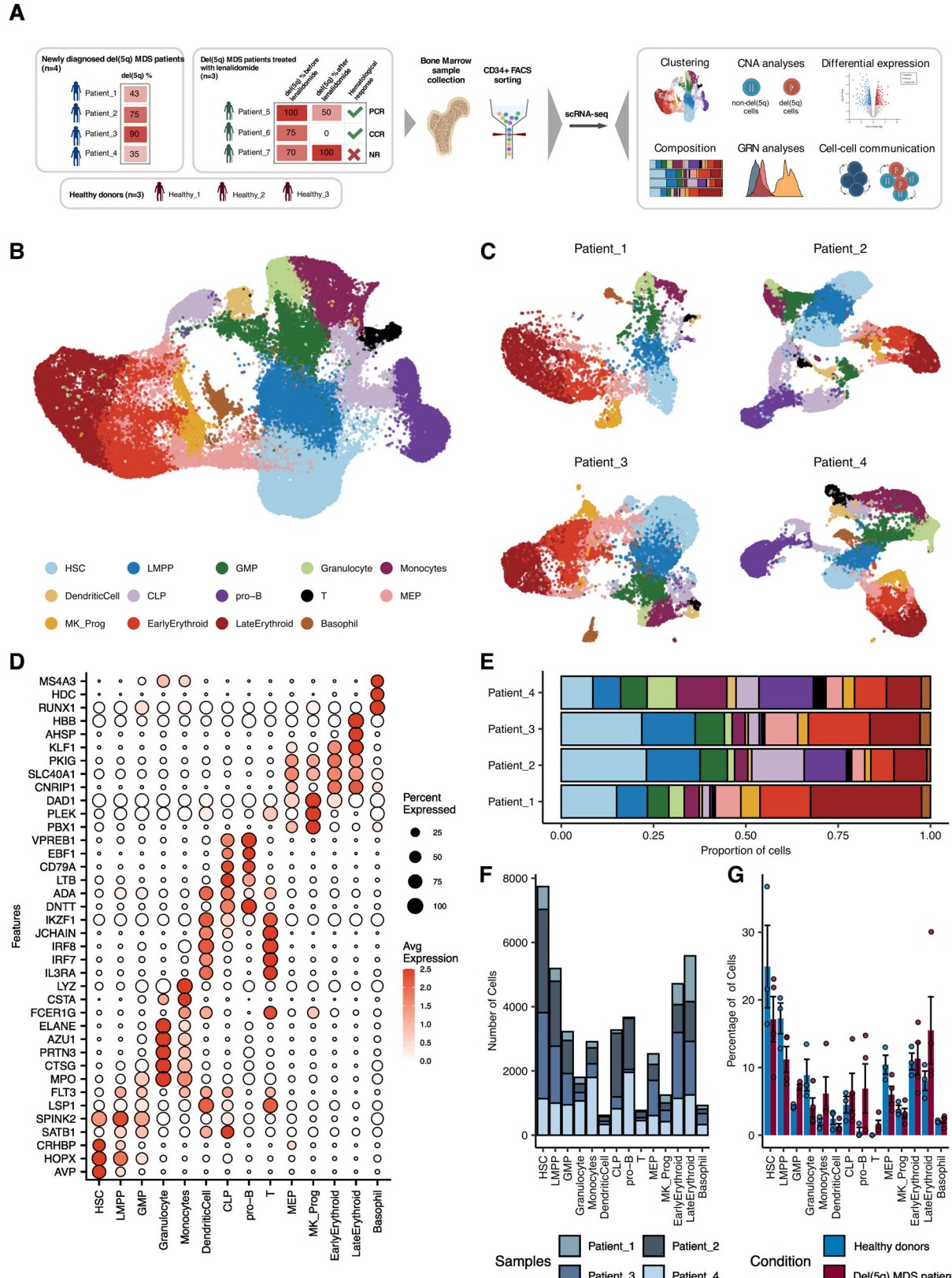

non-hematological tumors, such as *PRSS21*, which encodes for a tumor suppressor frequently hypermethylated in cancer[26], *MAP3K7CL*, whose downregulation serves as a biomarker in other types of cancer[27], and *CCL5*, whose downregulation is associated with high-risk MDS[28].

Due to the unexpected transcriptional similarity between del(5q) and non-del(5q) cells within MDS patients, we next performed a DE

analysis between del(5q) MDS cells and CD34⁺ cells from healthy donors. This comparison yielded 20 to 988 differentially expressed genes (FDR < 0.05 and |logFC|>2), depending on the progenitor cell (Supplementary Fig. 4A, Supplementary Data 2). Although most of these genes were cell-type-exclusive, they were enriched in similar pathways in most of the cell types (Fig. 4B, left panel). Genes

**Fig. 1 | Hematopoietic CD34⁺ cells from four independent del(5q) MDS patients were assayed by scRNAseq. A** CD34⁺ cells were obtained from bone marrow aspirates of newly diagnosed del(5q) MDS patients, healthy donors and patients treated with Lenalidomide, and were subjected to single-cell RNA sequencing and analysis. PCR partial cytogenetic responder, CCR complete cytogenetic responder, NR non-responder. Part of this figure was created with BioRender.com. **B** Uniform Manifold Approximation and Projection (UMAP) of 42,494 cells representing the expected 14 hematopoietic progenitors: HSC hematopoietic stem cells, LMPP lymphoid-primed multipotent progenitors, GMP granulocyte-monocyte progenitors; granulocyte progenitors; monocyte progenitors; dendritic cell progenitors; CLP common lymphoid progenitors; B-cell progenitors; T-cell progenitors; MEP megakaryocyte-erythroid progenitors; MK_Prog megakaryocyte progenitors; early erythroid progenitors; late erythroid progenitors; basophil progenitors. **C** Per patient UMAP showing the identity of the cells projected from the integrated space.

**D** Dotplot showing the percentage and value of the normalized expression of the canonical marker genes used to assign the cell identity to each cluster. **E** Barplot representing the contribution of cells from each patient to the different cell types. **F** Barplot representing the number of cells assigned to each cell type for the studied patients. **G** Barplot representing the percentage of cells assigned to each cell type for del(5q) MDS patients and healthy samples. $N = 7$ biologically independent samples were used ($n = 3$ healthy donors and 4 del(5q) MDS patients). Data are presented as mean values +/−SD. Two-sided Wilcoxon signed-rank test was used to calculate statistical significance. Exact $p$-values for the differential abundance of each hematopoietic progenitor between the del(5q) MDS and the healthy condition were the following: HSC: $p = 0.63$; LMPP: $p = 0.23$; GMP: $p = 0.06$; Granulocyte: $p = 0.23$; Monocytes: $p = 0.06$; DendriticCell: $p = 0.63$; CLP: $p = 1$; pro-B: $p = 0.23$; T: $p = 0.04$; MEP: $p = 0.11$; MK_Prog: $p = 0.63$; EarlyErythroid: $p = 1$; LateErythroid: $p = 0.23$; Basophil: $p = 0.4$.

overexpressed in del(5q) cells were enriched in cell cycle and mitosis-related signatures, such as DNA replication and mitotic nuclear division, and showed increased expression of DNA repair related genes, suggesting that loss of 5q confers increased proliferative potential. Additionally, del(5q) erythroid progenitors, LMPPs, GMPs, DCs, and monocyte progenitors showed significant upregulation of genes involved in the p53 signaling and, genes involved in the apoptosis pathway were significantly upregulated in LMPP, MEP and late erythroid progenitors, but not in early erythroid progenitor cells. Our results are in line with the increased levels of apoptosis described for del(5q) patients[29–31]. Downregulated genes showed enrichment in ribosomes and translation related pathways in all hematopoietic progenitors, in line with previous works that have described del(5q) MDS as a ribosomopathy[2,30,32]. Interestingly, besides the cytoplasmic translation, we also observed altered expression of genes associated with mitochondrial translation altered in HSCs, GMPs and granulocyte progenitors. The comparison of non-del(5q) and healthy cells resulted in 64-736 altered genes per progenitor (Supplementary Fig. 4B, Supplementary Data 2). Enriched processes were also homogeneous among most hematopoietic progenitors and, as expected, were similar to the ones observed in del(5q) vs healthy comparison (Fig. 4B, right panel).

Despite the low number of DE genes between del(5q) and non-del(5q) cells, we were interested in understanding whether differences in GRN might be observed between these two populations. Unlike DE analysis, which is performed in a gene-by-gene manner, GRN studies use data-driven grouping of genes to enable the identification of mechanistic transcriptional differences between conditions. Thus, we applied SimiC[33] to compute the regulatory activity of regulons and observed that although some regulons behaved uniformly (low regulatory dissimilarity score, in Supplementary Fig. 5A black-purple color) between the three conditions (del(5q), non-del(5q) and healthy cells), a group of regulons showed differential activity (high regulatory dissimilarity score, in Supplementary Fig. 5A yellow-orange color) across the conditions. Among them, three different regulon activity patterns arose. Firstly, a group of regulons that showed similar activity between non-del(5q) and del(5q) cells, and different to healthy cells, in line with DE analyses, such as the ones driven by *ZNF451*, *YBX1* and *PSPC1* (Fig. 4C). Secondly, there were regulons with differential activity between the three conditions, such as those driven by *JARID2*, *IRF1* and *KAT6B*, among others (Fig. 4D). The three regulons showed high activity in cells from healthy age-matched controls (61–84 years), whereas they presented a progressively lower activity in non-del(5q) cells, and their lowest activity in del(5q) cells. *JARID2* acts as a tumor suppressor and plays a crucial role in the leukemic transformation of myeloid neoplasms[34], and its deletion promotes an ineffective hematopoietic differentiation[35], suggesting that the low activity of this regulon may negatively impact the hematopoietic differentiation of these patients. *IRF1* is located in 5q31.1 and its deletion in one or both alleles has been observed in MDS and AML patients with chromosome 5

abnormalities[36]. *IRF1* has been described as a master HSC regulator, and its loss impairs HSC self-renewal and increases stress-induced cell cycle activation, suggesting that its low activity in patients could confer proliferative advantage[37]. Decreased expression of *KAT6B* in aged hematopoietic stem cells has been associated with impaired myeloid differentiation[38], suggesting that its almost non-existent activity in del(5q) cells may contribute to aberrant differentiation of these cells. Lastly, we detected regulons exhibiting differential activity between del(5q) and non-del(5q) cells and that showed no activity in healthy cells. In particular, regulons driven by *RERE* and *KDM2A* showed higher activity in del(5q) cells than in non-del(5q) cells (Fig. 4E). *RERE* negatively regulates the expression of target genes, and such genes are enriched in cytoplasmic translation, ribosome biogenesis and ribonucleoprotein complex biogenesis pathways, among others (Supplementary Fig. 5B). The *KDM2A* regulon was enriched in protein stabilization, regulation of cellular protein catabolic process and regulation of protein stability (Supplementary Fig. 5B). The association of *KDM2A* and ribosomal genes has been already described by previous studies, postulating that *KDM2A* overexpression reduces the transcription of rRNA[39,40].

Altogether, our results suggest a low transcriptional impact of 5q loss, with del(5q) and non-del(5q) cells presenting very similar gene expression alterations when compared to healthy controls, with such alterations being involved in processes that could contribute to abnormal hematopoietic differentiation. Nevertheless, although limited in number, genes and regulons specifically altered in del(5q) cells, such as those driven by *JARID2*, *KAT6B*, *RERE* or *KDM2A*, seem to be relevant for proliferation and myeloid differentiation, supporting the concept that cells harboring the deletion may have a more prominent role in the promotion of altered hematopoiesis.

## Abnormal cell-to-cell communication in del(5q) progenitors

To investigate whether the 5q deletion has a detrimental effect on cell-cell interactions between CD34⁺ progenitors, thus contributing to disease development, we performed a cell-to-cell communication analysis using Liana[41] in both del(5q) and healthy controls datasets. We identified 4,534 interactions in healthy controls, and 314 interactions that were common to all del(5q) MDS patients, most of them overlapping with those found in healthy cells (Fig. 5A). Despite this strong overlap, several differences between del(5q) MDS and healthy individuals were detected: in patients, monocyte progenitors were the most communicative cells, interacting mainly with early erythroid progenitors (Fig. 5B). However, in healthy donors, HSCs, GMPs, DC, monocyte and granulocyte progenitors were the most interactive compartments, with a notable communicative pattern between granulocyte and GMP/DC progenitors (Fig. 5C). Furthermore, genes involved in these differential interactions were overrepresented in different biological processes in each phenotype. For instance, interactions driven by healthy hematopoietic progenitors were enriched in negative regulation of apoptosis, HSC proliferation, leukocyte/DC differentiation, and

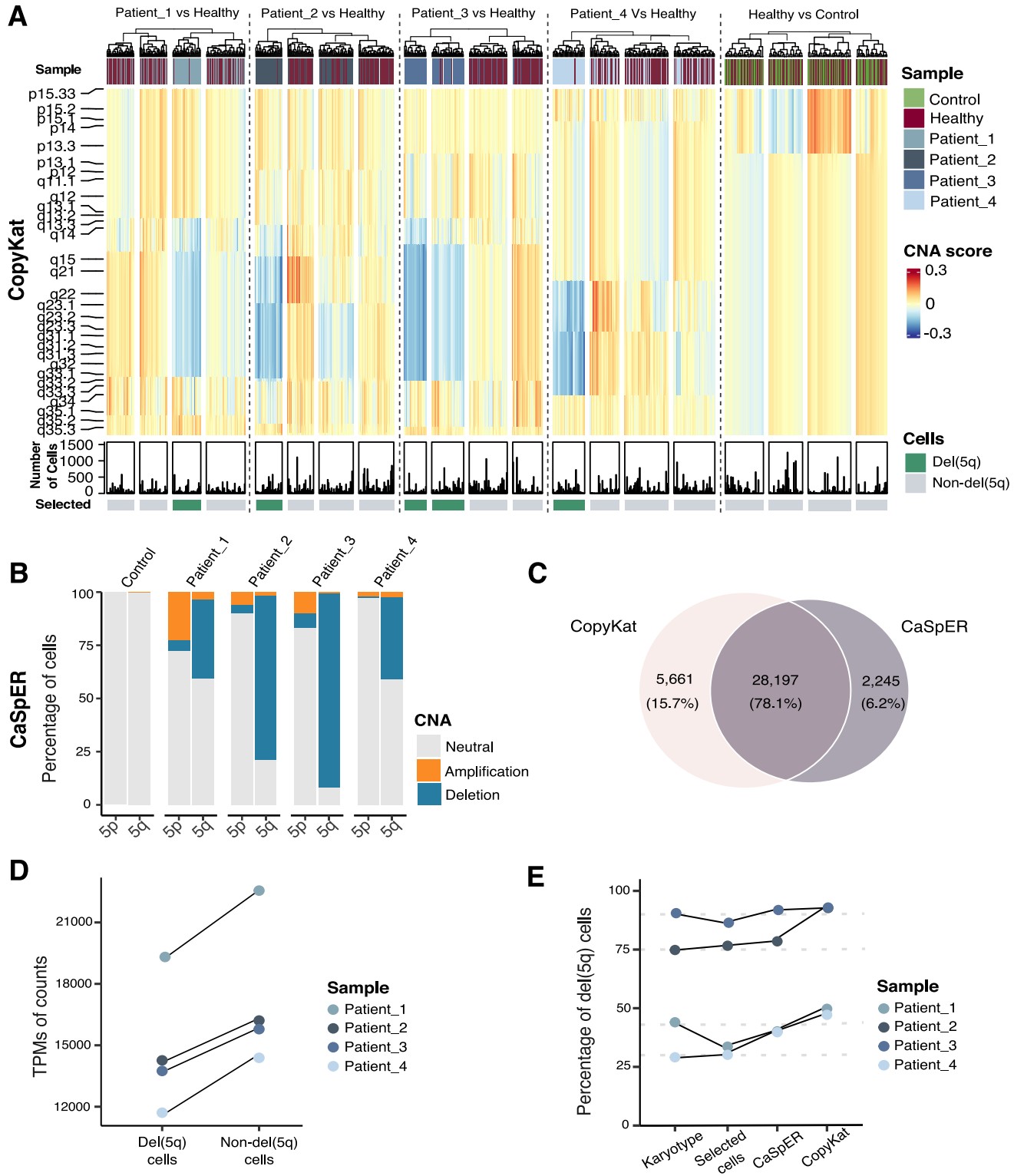

**Fig. 2 | Identification of cells harboring del(5q) deletion in MDS patients.**
**A** Heatmap of the results of CopyKat showing the copy number alteration score given to each 200 kb bins in chromosome 5. In order to represent cells, a clustering has been performed within each sample (kmeans with k = 80), and a posterior clustering has been applied to detect the clusters containing the cells harboring the deletion. The control sample used by the algorithm is an MDS sample with normal karyotype, while the healthy sample with normal karyotype represents an additional negative control for the analysis. **B** Barplot representing the percentage of cells inferred by CaSpER that harbor an amplification, a deletion or a normal number of copy number variation in each branch of chromosome 5 per patient. The control corresponds to an MDS sample with normal karyotype, which is used as a reference by the algorithm. **C** Venn diagram representing the number and percentage of cells classified as del(5q) by both algorithms. **D** Pseudobulk normalized expression of the 6 CDR-genes with higher expression in our dataset (*CD74, RPS14, BTF3, COX7C, HINT1* and *RPS23*) separated by genotype. *N* = 4 biologically independent samples were used. The number of del(5q) and non-del(5q) cells were used to generate the pseudobulks for each patient can be found in the Source Data. **E** Graph depicting the percentages of del(5q) cells inferred by karyotype, CaSpER and CopyKat for each patient. Selected cells correspond to the cells classified as del(5q) cells by both computational algorithms.

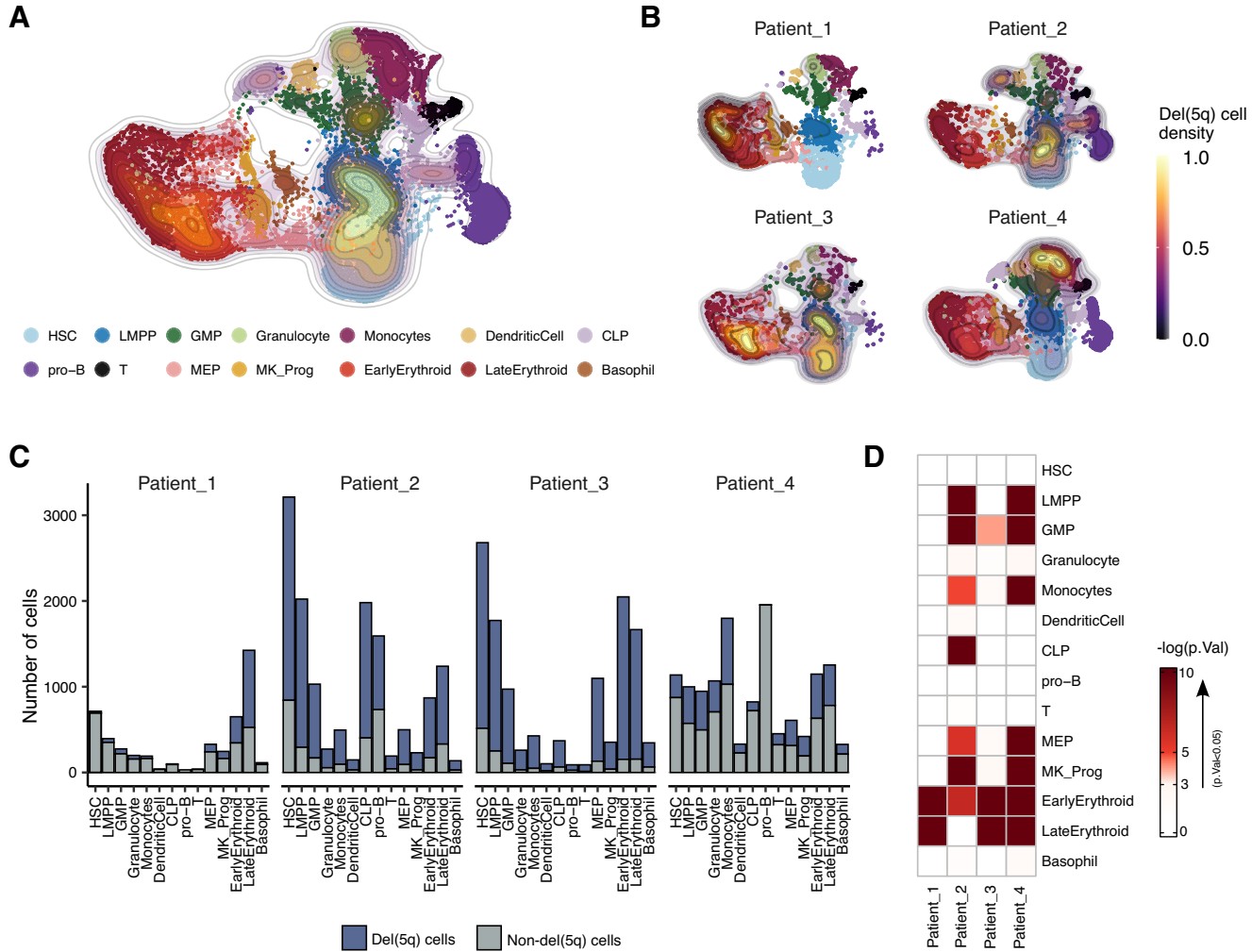

**Fig. 3 | Distribution of del(5q) cells within the CD34⁺ compartment of MDS patients. A** UMAP representing all the MDS samples integrated and colored by cell type. The density map represents the distribution of the cells classified as del(5q) by the two algorithms. **B** UMAPs with density maps representing the distribution of del(5q) cells per individual patient. **C** Barplots showing the number of del(5q) and non-del(5q) cells composing each cell type for each MDS patient. **D** Heatmap representing the enrichment of del(5q) cells (-log10($p$-value)) in each cell type. Any color different from white represents a statistically significant enrichment of del(5q) cells ($p$-value < 0.05). $p$-values were calculated using the one-sided hypergeometric test. The number of biologically independent replicates (cells) used for the hypergeometric test and the exact enrichment $p$-values can be found in the Source Data.

hemopoiesis, whereas those found in MDS progenitors were enriched in negative regulation of translation, oncogenic MAPK signaling and HIF-1 signaling (Fig. 5D). Focusing on interactions driven by del(5q) and non-del(5q) cells within the patients (Fig. 5B), we observed very subtle differences regarding the communicational pattern and the number of interactions observed for each of the compartments, and there were no interactions specifically established between del(5q) cells, corroborating the high similarity already described between del(5q) and non-del(5q) cells. Overall, our results are consistent with the previously described lack of significant differences in gene expression between del(5q) and non-del(5q) cells, suggesting that deregulation of hematopoiesis in patients with 5q MDS affects all CD34⁺ cells.

To uncover specific interactions that may contribute to the disease, we next focused on those interactions that had been gained or lost in MDS versus controls. There were 17 interactions identified in patients that were totally absent in healthy individuals, suggesting that additional communications arise when developing the disease. For instance, *AGTRAP* expressed in monocyte and late erythroid progenitors interacted with *RACK1* in HSCs, LMPPs, MEPs, pro-B and basophil progenitors (Fig. 5E). *AGTRAP* is known to be implicated in hematopoietic cell proliferation and survival[42], whereas *RACK1* has been postulated as a potential therapeutic target for promoting proliferation in

other myeloid neoplasms[43,44]. The fact that these molecules are highly expressed in MDS could potentially be contributing to the enhanced proliferation observed in MDS cells. In contrast, there were 37 interactions that appeared in the healthy donors and were absent in the patients, including the one established between *HMGB1* expressed in CLPs, DC, granulocyte, basophil, megakaryocyte, early erythroid and late erythroid progenitors, and *CXCR4* present in HSCs (Fig. 5F). *HMGB1-CXCR4* interaction is known to trigger the recruitment and activation of inflammatory cells in tissue regeneration[45,46], thus its loss could have a negative impact on the bone marrow niche. In summary, these analyses may allow the identification of potential interactions implicated in the pathogenesis of the disease that could represent new therapeutic targets.

### Effect of Lenalidomide treatment on the transcriptional programs of del(5q) and non-del(5q) cells from MDS patients

We next aimed to understand the effect of treatment with the standard-of-care, lenalidomide, on the transcriptional alterations observed in del(5q) and non-del(5q) cells. We performed scRNA-seq on CD34⁺ cells of two patients (Patient_5-6), which had achieved hematological response (one with partial cytogenetic response (PCR), and the other one with complete cytogenetic response (CCR), respectively) (clinical

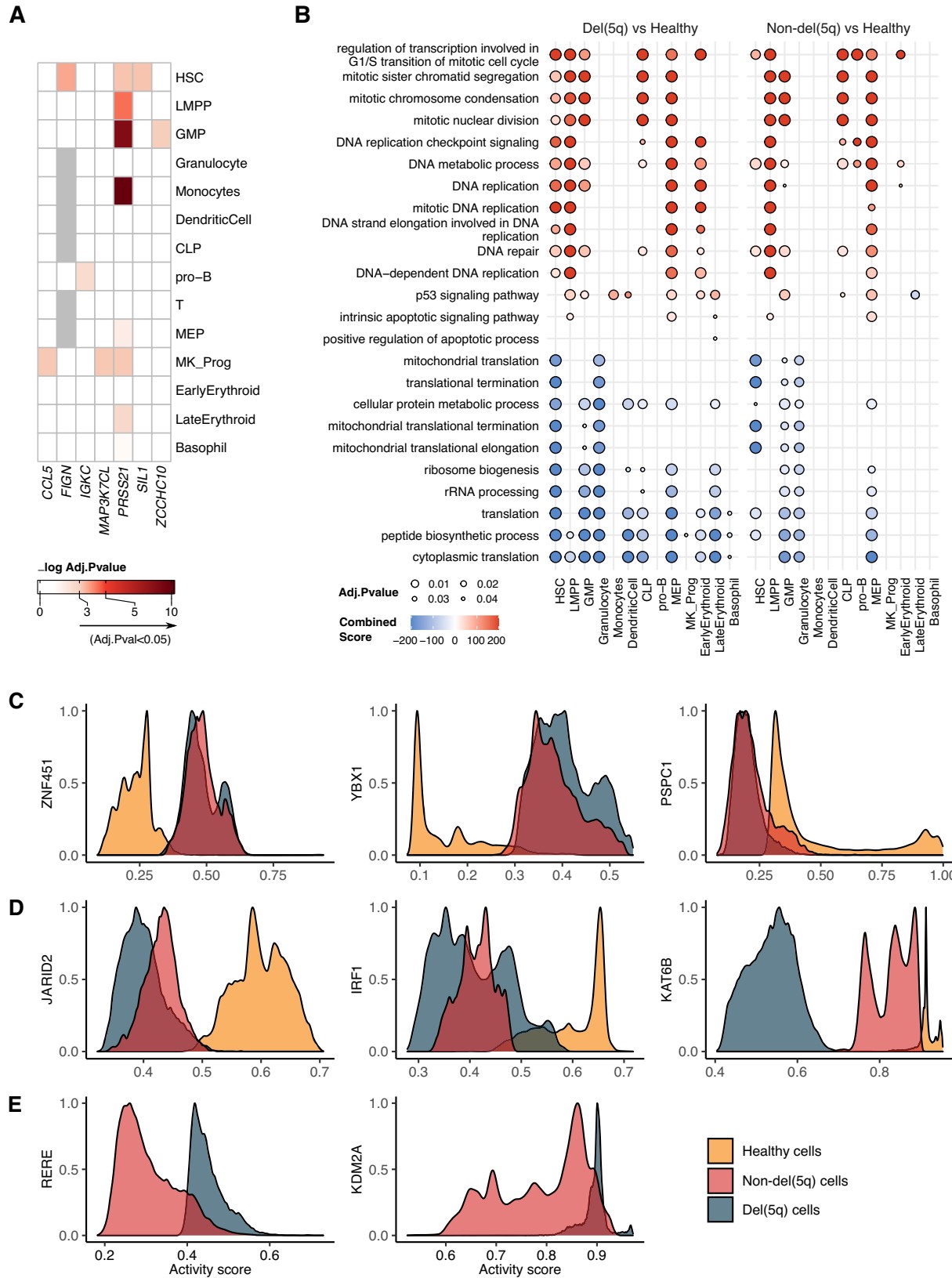

information in Supplementary Table 1). Data were integrated, clustered, manually annotated, and del(5q) cells were identified as described before (Fig. 6A, B). Patients showed different percentages of del(5q) cells which were consistent with karyotype results (Fig. 6C): the patient with PCR showed 1939 cells with del(5q) (37.13%), whereas the patient showing CCR presented only 11 cells with del(5q) after

treatment (0.15%), validating the persistence of del(5q) progenitor cells at the time of complete clinical and cytogenetic remission[9]. Similar to what we observed at diagnosis, the distribution of del(5q) cells was heterogeneous among patients (Fig. 6A, B), and both responders exhibited a statistically significant del(5q) enrichment in GMPs and erythroid progenitors. Interestingly, patient with PCR also

**Fig. 4 | Differential expression analysis between del(5q) and non-del(5q) cells within MDS samples exposes transcriptional similarities. A** Heatmap representing the differentially expressed genes (Benjamini–Hochberg-adjusted *p*-values < 0.05 and |logFC|>2) between del(5q) and non-del(5q) cells within each hematopoietic progenitor. The heatmap was created by combining *n* = 4 del(5q) MDS patients and generating pseudobulks per cell type. The two-sided edgeR's Likelihood Ratio Test was used to calculate *p*-values. The exact number of biologically independent replicates (del(5q) and non-del(5q) progenitor cells), as well as the specific *p*-values for each differentially expressed gene can be found in the Source Data. **B** Dotplot representing statistically significant biological processes and pathways (Benjamini–Hochberg-adjusted *p*-values < 0.05) for differentially

expressed genes obtained in del(5q) versus Healthy and the non-del(5q) versus Healthy contrasts. The one-sided hypergeometric test was used to calculate *p*-values. Del(5q) and non-del(5q) cells were derived from *n* = 4 del(5q) MDS patients, whereas healthy cells were derived from *n* = 3 healthy donors. Biologically *inde*-pendent replicates (del(5q), non-del(5q) and healthy progenitor cells) used for each comparison are specified in the Source Data. **C–E** Histograms representing the activity score in all the cells separated by conditions. Some regulons behaved similarly in the MDS samples (non-del(5q) and del(5q) cells) compared to healthy cells (**C**), while other regulons behaved differently in the three different conditions (**D**). Some inferred regulons had an activity score on the MDS samples, while lacking on the healthy samples (**E**).

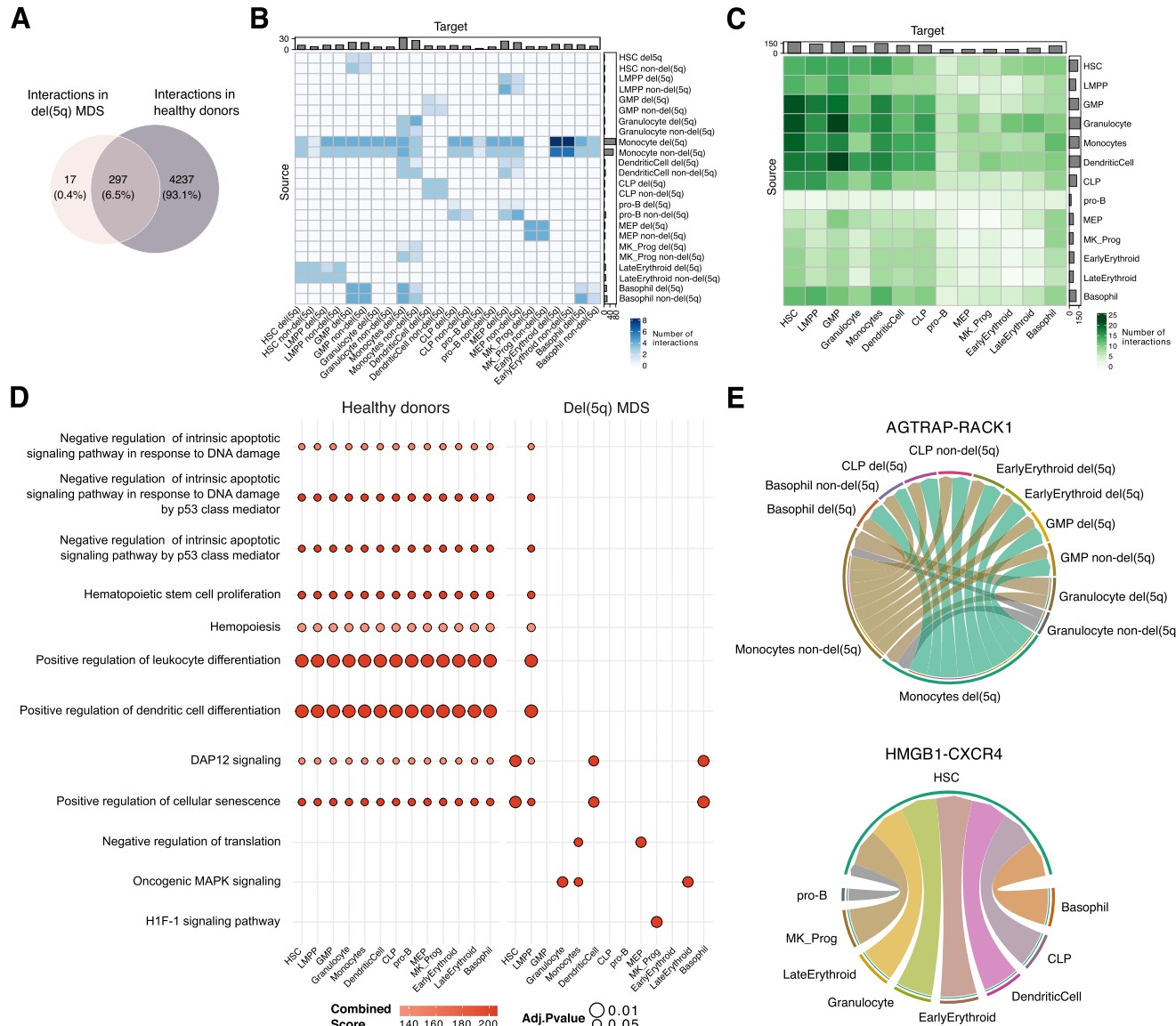

**Fig. 5 | Cell-to-cell communication analysis reveals shared and unique interactions in del(5q), non-del(5q) and healthy cells. A** Venn diagram showing the number of unique interactions in del(5q) MDS and healthy samples. Healthy unique interactions were considered as those present in at least one of the healthy individuals, while MDS unique interactions were those that were present in all the patients. Interactions were inferred from *n* = 4 del(5q) MDS patients and *n* = 3 healthy donors. **B** Heatmap depicting the number of interactions triggered by del(5q) and non-del(5q) MDS cells, **C** as well as those established among healthy hematopoietic progenitors. The Source represents the cell types that express the

ligand, whereas the Target represents the cells that express the receptor. **D** Dotplot representing statistically significant biological processes and pathways (Benjamini–Hochberg-adjusted *p*-value < 0.05) in which are enriched the encoding genes taking part in the healthy and MDS interactions. The one-sided hypergeometric test was used to calculate *p*-values, whose exact values can be found in the Source Data. **E** Chord diagram representing the unique MDS interaction AGTRAP-RACK1 among different del(5q) and non-del(5q) progenitors. **F** Chord diagram depicting the unique healthy interaction HMGB1-CXCR4 established by healthy hematopoietic progenitors.

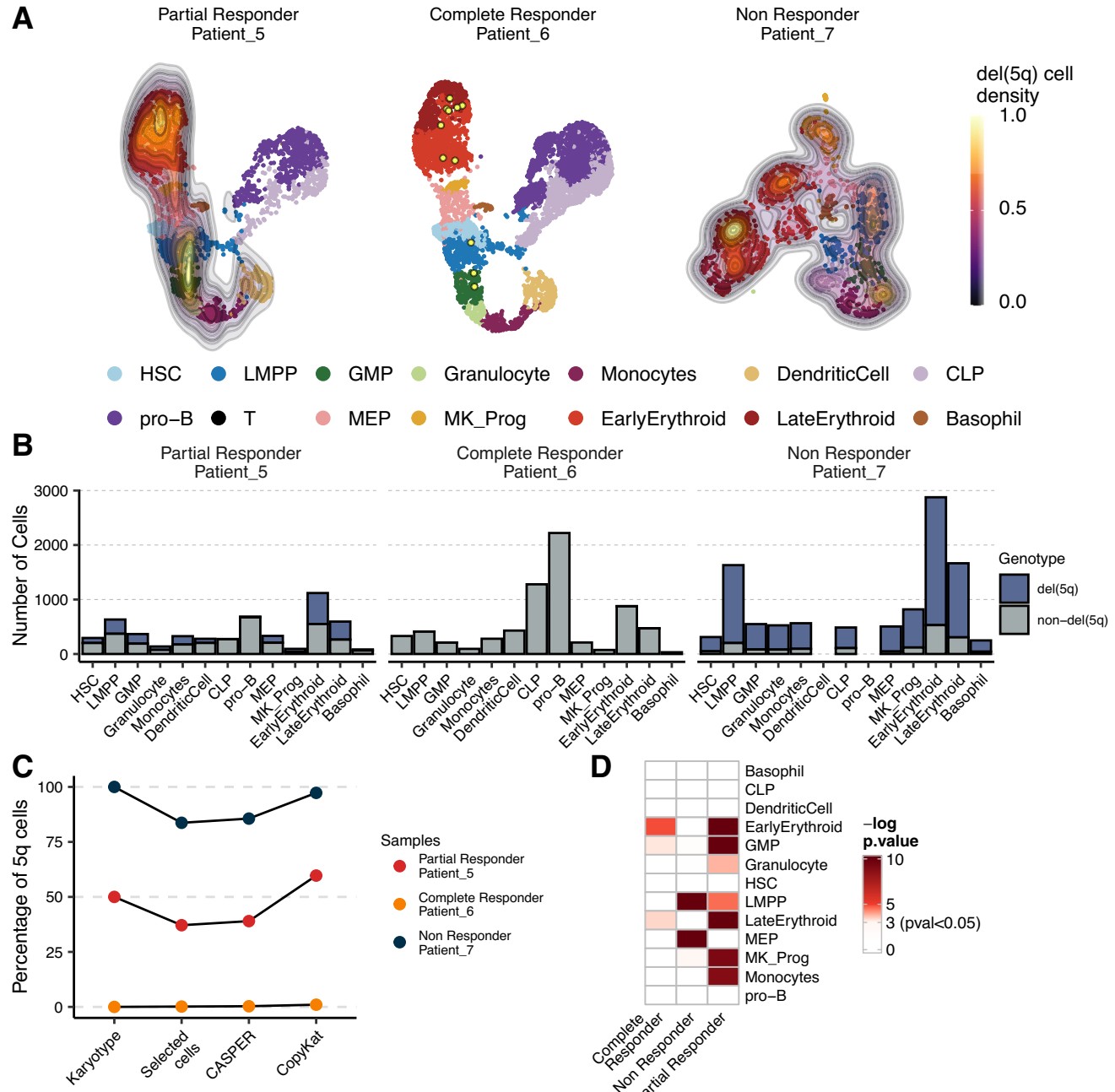

**Fig. 6 | Distribution of del(5q) cells within CD34⁺ progenitors after lenalido-mide treatment. A** UMAP depicting the del(5q) density across the different hematopoietic progenitors obtained in three patients after lenalidomide treatment. HSC hematopoietic stem cells, LMPP lymphoid-primed multipotent progenitors, GMP granulocyte-monocyte progenitors; granulocyte progenitors; monocyte progenitors; dendritic cell progenitors, CLP common lymphoid progenitors; B-cell progenitors; T-cell progenitors, MEP megakaryocyte-erythroid progenitors, MK_Prog megakaryocyte progenitors; early erythroid progenitors; late erythroid progenitors; basophil progenitors. **B** Barplots showing the number of

del(5q) and non-del(5q) cells composing each cell type for each MDS patient. **C** Percentage of the cells identified as del(5q) by karyotype, CASPER, CopyKat, and the selection by intersecting the two algorithms. **D** Heatmap representing the enrichment of del(5q) cells (log10(*p*-value)) in each cell type. Any color different from white represents a statistically significant enrichment of del(5q) cells (*p*-value < 0.05). *P*-values were calculated using the one-sided hypergeometric test. The number of biologically independent replicates (cells) used for the hypergeometric test and the exact enrichment *p*-values can be found in the Source Data.

showed an enrichment in LMPPs, megakaryocyte, monocyte, and granulocyte progenitors (Fig. 6D).

We have demonstrated in the previous analyses that at diagnosis both del(5q) and non-del(5q) progenitors displayed transcriptional profiles linked to an aberrant hematopoiesis. Since both PCR and CCR patients were in hematological response, we hypothesized that the remaining CD34⁺ cells after lenalidomide treatment, which are mainly composed of non-del(5q) progenitors, must be able to promote

improved hematopoiesis and thus restore the transcriptional profile of normal progenitor cells. To demonstrate that lenalidomide, besides the potential apoptosis of del(5q) cells, could reverse transcriptional alterations harbored by non-del(5q) cells in responder patients, we performed a DE analysis between non-del(5q) cells from the CCR and PCR and the four patients at diagnosis. This comparison revealed significant transcriptional changes after lenalidomide treatment, resulting in 622–3609 genes in the different progenitor populations

for the CCR (Supplementary Fig. 4C, Supplementary Data 2) and between 458–2409 genes for the PCR (FDR < 0.05 and |logFC|>2) (Supplementary Fig. 4D, Supplementary Data 2). Note that these transcriptional differences are significantly greater than those related to patient heterogeneity at diagnosis (see previous sections), indicating that most uncovered altered genes after treatment are probably due to treatment effect rather than patient heterogeneity. Genes altered upon treatment were enriched in ubiquitination and proteasome-mediated catabolic processes, and in phosphatidylinositol related pathways, which is in line with the mechanism of action described for lenalidomide in MDS patients[47,48]. Moreover, we detected an enrichment in autophagy-related processes. Overall, our results suggested an increase of the two most important protein degradation pathways in non-del(5q) cells upon lenalidomide treatment (Fig. 7A, first and second panels, Supplementary Table 2). Furthermore, hematopoietic progenitors exhibited an increased expression of genes involved in erythroid differentiation and erythropoietin signaling after treatment (Fig. 7B), validating the enhanced erythropoiesis in response to treatment[19]. Our analyses also detected a positive enrichment of PD-L1 expression and PD-1 checkpoint in non-del(5q) cells of the responder patients after treatment, suggesting a potential immunosuppressive mechanism of these cells in response to lenalidomide (Fig. 7A, first and second panels, Supplementary Table 2).

GRN analyses evidenced that some of the alterations described at diagnosis were potentially reverted after treatment in non-del(5q) cells. IRF1, the master HSC regulator located in 5q31.1, which showed abnormally low activity at diagnosis in non-del(5q) cells, showed an increased activity after treatment in both patients, with the PCR not reaching the activity level seen for healthy cells, and the CCR showing an augmented activity comparable to the healthy cells (Fig. 7C). KAT6B, whose lower expression has been associated with impaired myeloid differentiation, showed an augmented activity in both patients despite not reaching the activity level of healthy cells. Finally, CUX1[49], a TF frequently mutated in myeloid malignancies and whose knockdown leads to an MDS-like phenotype, presented similar activity in non-del(5q) cells, showing higher activity than at diagnosis (Fig. 7C).

Importantly, although some transcriptional lesions were reverted upon lenalidomide treatment, non-del(5q) cells continue exhibiting altered expression of ribosome-related genes, showing a negative enrichment of processes related to ribosomes, translation, and mitochondrial translation when compared to healthy cells (Fig. 7A, fourth and fifth panels, Supplementary Fig. 4E, F, Supplementary Data 2 and Supplementary Table 2). After treatment, early and late erythroid non-del(5q) progenitors from responding patients showed no statistically significant changes in these pathways. Moreover, GRN analyses detected groups of regulons with similar activity for non-del(5q) cells at diagnosis and after treatment response, but with a different activity to the healthy cells, indicating that lenalidomide did not affect their aberrant activity. Some examples included the tumor suppressor JARID2[34,35], ZNF451, a TF whose high expression in leukemic cells has been associated with poor outcome[50], and NCOR1, a regulator of erythroid differentiation[51] (Fig. 7C). Moreover, non-del(5q) cells exhibited, both at diagnosis and after treatment, abnormal high activity of two regulons that were not active in healthy cells: ADNP and SMARCE1 (Fig. 7C). Globally, these results indicate that treatment with lenalidomide has the potential to revert some of the transcriptional alterations present at diagnosis in non-del(5q) cells at least in patients that responded to lenalidomide. Nevertheless, some of the transcriptional alterations present at diagnosis were not modified which could be relevant for abnormal hematopoiesis, and potentially, for the future relapse of the patients.

In line with what has been observed in non-del(5q) cells from the PCR, the remaining del(5q) cells generally exhibited significant upregulation of genes involved in ubiquitin and phosphatidylinositol signaling and, autophagy and apoptosis pathways when compared to

del(5q) cells at diagnosis (Supplementary Figs. 4G, 6A, Supplementary Data 2 and Table 3), which is consistent with the mechanism of action of lenalidomide[47,48]. However, these cells showed reduced expression of genes implicated in ribosomal and mitochondrial translation compared to diagnosis, along with diminished expression of DNA repair associated genes. (Supplementary Fig. 6A). This suggests that lenalidomide does not fully reverse key transcriptional alterations that may underlie the ribosomopathy characterizing the disease.

## Lenalidomide does not correct transcriptional alterations of del(5q) cells of a refractory MDS patient

Finally, to understand the transcriptional alterations associated with a lack of hematological response after lenalidomide treatment, we performed scRNAseq on CD34+ cells of an additional patient (Patient_7), who was refractory to lenalidomide (non-responder, NR). Data were processed as described previously (clinical information in Supplementary Table 1), showing 83.8% of del(5q) cells (Fig. 6A–C), with a statistically significant increased abundance in LMPPs, MEPs and megakaryocyte progenitors (Fig. 6D). We then analyzed the transcriptional differences between the remaining del(5q) cells of the responder that presented PCR, and those of the NR patient. This analysis identified 116–2244 differentially expressed genes (FDR < 0.05) per progenitor (Supplementary Fig. 4H, Supplementary Data 2). Del(5q) cells from the patient in PCR showed statistically significant enrichment in processes and pathways related to protein ubiquitination, proteasomal protein catabolic process, phosphatidylinositol and autophagosome when compared to the NR. Moreover, these cells also exhibited an increased expression of genes involved in erythropoietin signaling when compared to the cells from the NR (Fig. 7A, third panel). Interestingly, these processes are similar to the ones detected for non-del(5q) cells when comparing these cells to those at diagnosis (see previous section), and have been described as a lenalidomide response in non-del(5q) MDS patients[47,48]. The remaining del(5q) cells from the patient in PCR also exhibited enrichment of PD-L1 expression and PD-1 checkpoint pathway when compared to the refractory patient. These analyses suggested low transcriptional alterations promoted by lenalidomide treatment in the NR patient. Accordingly, DE analysis of del(5q) cells at diagnosis and after treatment in the NR patient yielded 20–121 differentially expressed genes per hematopoietic progenitor (Supplementary Fig. 4I, Supplementary Data 2). These few differences resulted in subtle changes in protein ubiquitination and cell cycle-related processes after treatment (Supplementary Fig. 6B, Supplementary Table 4), showcasing that lenalidomide did not have a high transcriptional impact on del(5q) cells of the NR.

GRN analysis demonstrated a large number of regulons that showed changes in activity after treatment in del(5q) cells from the patient in the PCR but not in the refractory patient. For example, regulons driven by IRF1, JARID2, NCOR1, and CUX1, which showed aberrant low activity at diagnosis that was partially recovered upon treatment, presented very reduced activity in the NR patient, which was lower than that observed in the PCR, and at diagnosis (Fig. 7D). Collectively, these results suggest that in NR patients, lenalidomide treatment is not able to reverse part of the transcriptional lesions carried by (5q) cells, which seems to be associated with the lack of hematological response.

## Discussion

Establishing the relationship between genomic and transcriptional abnormalities in hematological malignancies has allowed researchers to characterize the molecular pathogenesis of diseases such as MDS and to identify new potential targets[10–14,18]. However, for the most part, these studies have been performed using bulk sequencing data, which precludes a direct association on a per cell basis between genomic and transcriptomic alterations. Using scRNAseq data from CD34+ cells from patients with del(5q) MDS, we have been able to identify cells

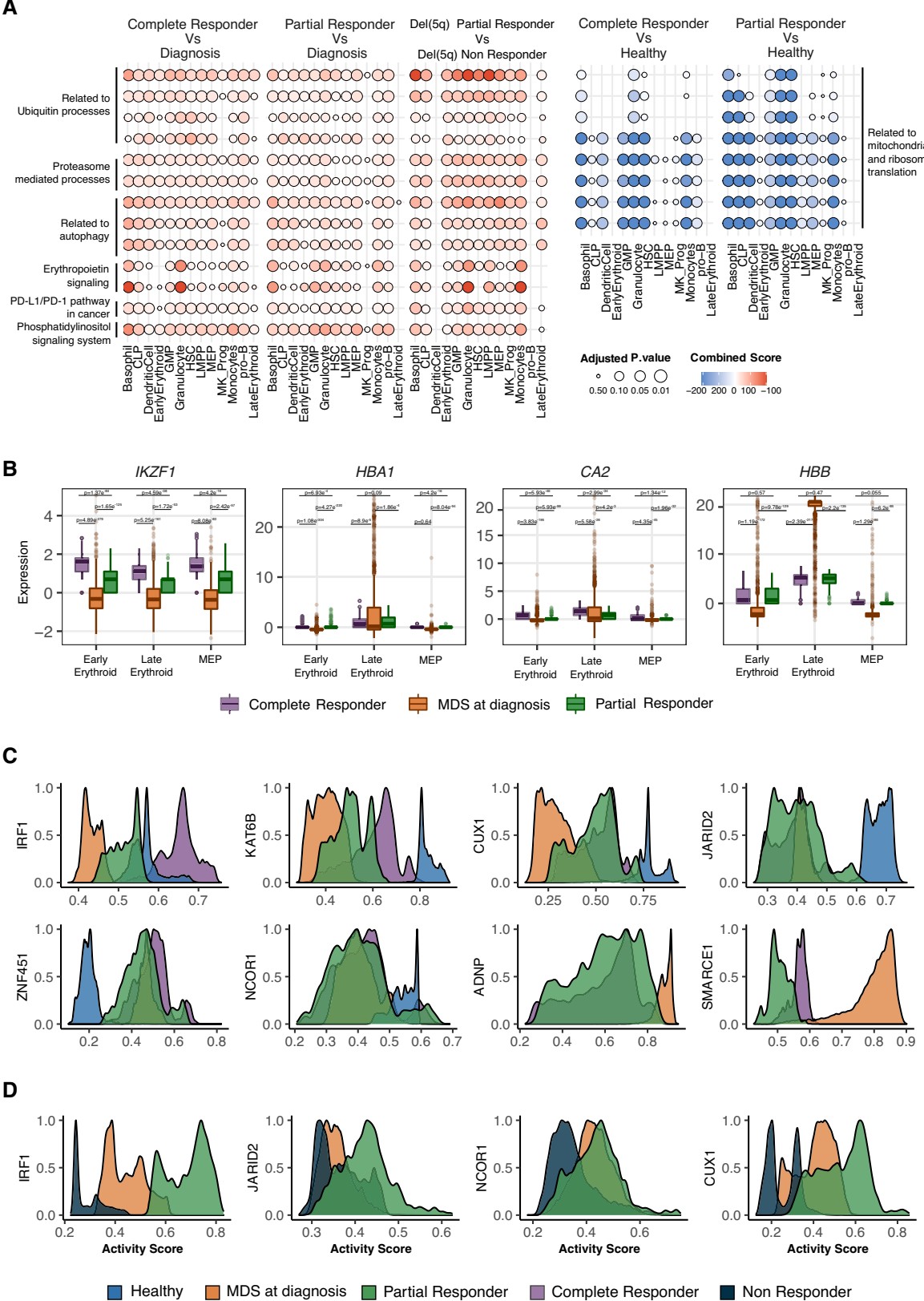

with del(5q) and non-del(5q), which enabled us to compare the transcriptional profile and GRNs of both populations within the same patient, thereby mitigating potential confounding factors associated with interindividual variability and microenvironment influences, and thus made strides in the understanding of the molecular pathogenesis of del(5q) MDS.

Single cell RNA-seq coupled with CNA detection algorithms offers the opportunity to link the genetic information to the transcriptional profile of each individual cell. Nonetheless, this type of analyses remain challenging due to the sparse and noisy nature of single-cell data[52]. To ensure robust findings we have employed two CNA detection methods, one based on transcript abundance[23] and another utilizing allelic

**Fig. 7 | Differential expression between treated and untreated patients unravels persistent transcriptional alterations after lenalidomide treatment.**
**A** Dotplot representing statistically significant biological processes and pathways (Benjamini–Hochberg-adjusted $p$-value < 0.05 and |logFC|>2) for differentially expressed genes obtained in different comparisons: non-del(5q) cells of the complete responder vs at diagnosis (1st panel); non-del(5q) cells of the partial responder vs at diagnosis (2nd panel); del(5q) cells of the partial responder vs the non-responder (3rd panel); non-del(5q) cells of the complete responder vs healthy cells (4th panel); non-del(5q) cells of the partial responder vs healthy cells (5th panel). For $p$-value calculation, one-sided hypergeometric test was used. Specific $p$-values for statistically significant biological processes can be found in the Source Data. The detailed breakdown of the grouped processes shown can be found in Supplementary Table 2. **B** Boxplot showing the normalized expression of erythroid differentiation-related genes for non-del(5q) cells in MDS at diagnosis ($n = 4$) or after

treatment with lenalidomide (partial responder, $n = 1$; complete responder, $n = 1$). Biologically independent replicates (cells) for hematopoietic progenitors were: Early Erythroid: $n = 5196$ (MDS at diagnosis); $n = 2200$ (Partial Responder); $n = 3496$ (Complete Responder); Late Erythroid: $n = 7168$ (MDS at diagnosis); $n = 1056$ (Partial Responder); $n = 1880$ (Complete Responder); MEP: $n = 3108$ (MDS at diagnosis); $n = 824$ (Partial Responder); $n = 844$ (Complete Responder). Two-sided Wilcoxon signed-rank test was used to calculate $p$-values, that were then Benjamini–Hochberg-adjusted. Box plots indicate median (middle line), 25th, 75th percentile (box) and 5th and 95th percentile (whiskers) as well as outliers (single points). Exact $p$-values are shown within the figure. **C** Graphs representing the activity scores of proliferation and differentiation-associated transcription factors in healthy cells and non-del(5q) cells of MDS patients at diagnosis and after lenalidomide treatment, as well as **D** in del(5q) cells of MDS patients at diagnosis, with a partial response and with no response to lenalidomide. Specific activity scores can be found in the Source Data.

imbalance in heterozygous SNPs[24]. While expression-based approaches operate under the assumption that amplifications or deletions generally result in up- or down-regulation of genes within the affected region of the genome, allele-based methodologies focus on analyzing deviations in heterozygous allele frequency. We acknowledge the potential for minor misclassification due to the inherent complexity of applying CNA detection algorithms to scRNA-seq data, however, we believe that this would be minimal, as clearly demonstrated in the results section (Supplementary Fig. 2). Our confidence in employing two complementary methods stems from the belief that this approach enhances classification accuracy and mitigates false-positive results arising from local expression variations unrelated to genomic copy numbers, as well as from data sparsity and allele-specific transcriptional stochasticity[23,24].

The stem-cell origin of del(5q) MDS has been previously demonstrated[9,53] as well as the presence of del(5q) cells even at the stage of cytogenetic response[9]. By being able to dissect the presence of del(5q) in individual cells and identifying the different types of progenitors, our study suggests uneven distribution of del(5q) in different progenitor cell populations, and also an heterogeneous distribution according to the patients. Although we cannot establish a correlation between the phenotype of the disease, the clinical symptoms, and the specific distribution of del(5q), we were able to identify an enrichment of del(5q) cells in GMP, megakaryocyte and erythroid progenitors in patients, not described to date. Future studies with larger cohorts of cases may uncover the nature of the detected enrichment and provide possible associations between the distribution of del(5q) cells and clinical characteristics of the patients.

The comparison between the transcriptional profiles of del(5q) cells and non-del(5q) within the same patient provides insights regarding the impact of 5q deletions on pathogenesis of del(5q) MDS. The presence of similar transcriptional alterations in both del(5q) and non-del(5q) suggests that in del(5q) MDS patients, both types of cells are implicated in the disease and contribute to the promotion of aberrant hematopoietic differentiation. Nevertheless, the boosted aberrant activity of some regulons in del(5q) cells, may exert a more prominent role of these cells in the aberrant hematopoiesis of these patients. Furthermore, this fact might indicate the presence of other additional factors, such as a disrupted microenvironment, as evidenced by recent studies elucidating its impact on the initiation and progression of MDS[54,55]. Moreover, the presence of additional shared genetic lesions between del(5q) and non-del(5q) cells may play a key role in the observed transcriptional similarities. Future studies with larger cohorts of cases may elucidate the underlying nature of these observed transcriptional similarities.

Specifically, del(5q) and non-del(5q) cells presented alterations in biological processes described in del(5q) MDS, including cell proliferation[56,57], p53 signaling pathway[14] and apoptosis. In this regard, previous murine models generated by the inactivation of *RPS14* described a p53 dependent apoptosis in the erythroid lineage between

basophilic/early chromatophilic erythroblasts to poly/orthochromatophilic erythroblasts[30], but our data suggests that it could already occur at the CD34+ progenitor level. Furthermore, in addition to the negative enrichment in ribosome and cytoplasmic translation processes identified in our data, which are consistent with previous studies[2,30,32], we observed an altered function of mitochondrial ribosomes in HSCs and granulocyte-lineage progenitors. Mitochondrial translation and mitoribosomes fulfill a pivotal function in cell cycle regulation and apoptosis signaling[58], and several works have found gene expression alterations in mitoribosome or associated proteins in relation to different cancers[59]. Hence, the altered mitochondrial translation in del(5q) cells could be contributing to the molecular pathogenesis of del(5q) MDS. Despite these findings, a deeper GRN analysis led to the characterization of a number of transcriptional differences between del(5q) and non-del(5q) cells, identifying a group of regulons with different activity between both genotypes. Some of the GRNs specifically altered in del(5q) cells, such as those driven by *JARID2*, *KAT6B*, *RERE* or *KDM2A*, seem to be relevant for proliferation and myeloid differentiation, supporting the concept that cells harboring the deletion may have a more prominent role in the promotion of altered hematopoiesis. Altogether, our data manifests a general transcriptional dysregulation in all the progenitor cells of del(5q) MDS patients, and shows that progenitors harboring the del(5q) deletion show additional alterations.

Lenalidomide represents the standard-of-care for transfusion-dependent del(5q) MDS patients. Our analyses suggest a close relationship between lenalidomide-mediated reversal of the transcriptional alterations harbored by hematopoietic progenitors and hematological response. In this sense, we observed that lenalidomide treatment alters the transcriptional profile of both del(5q) and non-del(5q) cells of responder patients, reverting part of the alterations observed in these cells. For example, the remaining non-del(5q) cells of responder patients show increased levels of proteasome-mediated catabolic and autophagia related processes after treatment, suggesting a compensatory increase of the two main mechanisms of intracellular protein degradation as a result of the treatment. Moreover, we detected a direct effect of lenalidomide in erythropoietin signaling, promoting the upregulation of genes involved in erythropoiesis, a mechanism that so far has been described for non-del(5q) MDS[60]. Nevertheless, certain alterations that were initially present at the time of diagnosis in del(5q) and non-del(5q) cells remain detectable at this stage of clinical evaluation, which could explain the relapses seen after the treatment. This association between reversal of transcriptional alterations and clinical response is further supported by the fact that both del(5q) and non-del(5q) cells of a non-responder patient presented very few transcriptional alterations upon lenalidomide treatment. Altogether, these results may be at odds with previous data suggesting that lenalidomide induces synthetic lethality to suppress the malignant clone without significant effect on the growth of cytogenetically normal CD34+ cells[61]. Our data demonstrate that

lenalidomide affects both del(5q) and non-del(5q) cells, and suggest that the hematological response of del(5q) MDS patients may be due to the correction of transcriptional alterations in both types of cells.

Finally, the characterization of the transcriptional profile of remaining cells after lenalidomide treatment could help to unveil novel therapeutic strategies that could be effective in the eradication of malignant cells. In this sense, our results showed an enrichment in *PD-1/PDL-1* pathway after the treatment in non-del(5q) cells which was not detected in the NR patient, suggesting a putative immune evasion mechanism of these cells. Interestingly, lenalidomide has shown increased cytotoxic activity in combination with immune-checkpoint blockade in multiple myeloma[62]. Thus, future studies will confirm if, as our results suggest, this combination could serve as a promising therapeutic strategy for these patients.

## Methods

The research performed in this work complies with all relevant ethical regulations: the study was approved by the research ethics committee of University of Navarra, and informed consent was obtained from all patients and healthy donors.

### Sample collection

Bone marrow aspirates were obtained from healthy age-matched controls [($n = 3$), median age, 72 years, range, 61–84 years] and from patients with MDS [($n = 7$), median age, 84 years, range, 80–91 years] from the Clinica Universidad de Navarra and collaborating hospitals after the study was approved by the research ethics committee of University of Navarra, and informed consent was obtained. MDS patients and healthy donors were not economically compensated for the samples donated. Patient's data were fully anonymized, and all patients provided informed written consent for the use and publication of data from their medical records such as age, sex, and diagnosis for research purpose. Healthy controls were patients undergoing orthopedic surgery. As del(5q) MDS is a subtype of MDS which is predominantly found in females, and these samples are not very abundant, in the present study we included all the patients for which we were able to obtain samples. Thus, we were not able to select patients by sex or to carry out a sex or gender analysis. The patients' clinical characteristics are shown in Supplementary Table 1.

### Fluorescence-activated cell sorting

For purification of CD34$^+$ cells, BM samples were lysed in 1X of BULK lysis buffer (150 mM NH4Cl, 10 mM CHKO3 and 0.1 mM EDTA in deionized water) for 15 min at a sample to bulk ratio of 1:10, and centrifuged for 5 min at 500 × $g$ to eliminate red blood cells. Next, cells were stained using CD34-APC (clone 581; Beckman Coulter, #IM2472, lot number 200504, dilution 5:100) and CD45-PerCPCy5.5 (clone HI30; Biolegend, #304028, lot number 200504, dilution 1:100) for 15 min at RT. CD34$^+$ CD45$^+$ cells were then sorted in a BD FACSAria II (BD Biosciences) and directly used for scRNA seq analysis. BD FACSDIVA v8 software was used for flow cytometry data analysis.

### scRNA-seq library preparation

The transcriptome of the bone marrow CD34$^+$ cells were examined using NEXTGEM Single Cell 3′ Reagent Kits v3 and v3.1 (10X Genomics) according to the manufacturer's instructions. Between 5000 and 17,000 cells, depending on the donor, were loaded at a concentration of 700–1200 cells/μL onto a Chromium Controller instrument (10× Genomics) to generate single-cell gel bead-in-emulsions (GEMs). In this step, each cell was encapsulated with primers containing a fixed Illumina Read 1 sequence, a cell-identifying 16-bp 10× barcode, a unique molecular identifier (UMI), and a poly-dT sequence. Upon cell lysis, reverse transcription yielded full-length, barcoded cDNA, which was then released from the GEMs, amplified using polymerase chain reaction, and purified using magnetic beads (SPRIselect, Beckman

Coulter). Enzymatic fragmentation and size selection were used to optimize the cDNA size prior to library construction. Fragmented cDNA was then end-repaired, A-tailed, and ligated to Illumina adapters. A final polymerase chain reaction amplification using barcoded primers was performed for sample indexing. Library quality control and quantification were performed using a Qubit 3.0 Fluorometer (Life Technologies) and an Agilent 4200 TapeStation System (Agilent), respectively. Sequencing was performed on NextSeq500 and Next-Seq2000 instruments (Illumina) at an average depth of 30,000 reads/cell.

### Single-cell RNA-seq analysis

The scRNA-seq data was demultiplexed and aligned to the human reference genome (GRCh38). The feature-barcode matrix was quantified using Cell Ranger (v6.0.1) from 10X Genomics. Computational analysis was carried out using Seurat[63] (v4.2.0). In order to remove the possible heterotypic and homotypic doublets/multiplets, cells were analyzed with scrublets[64] (v0.2.3) and removed. Cells underwent quality control filters based on the number of detected genes, number of unique molecular identifiers (UMIs), the cell complexity by calculating the ratio between the logarithm of the number of genes and the logarithm of number of UMIs, and the proportion of UMIs mapped to mitochondrial and ribosomal genes per cell. Each sample was analyzed searching for effects of cell cycle heterogeneity by calculating cell cycle phase scores based on canonical markers, without needing to apply any correction to the expression of the cells. Thus, each dataset was normalized, 2000 highly variable genes were identified, and unwanted sources of variation were removed. Integration of all datasets was performed using Seurat's canonical correlation analysis. Samples from del(5q) MDS patients and healthy donors were integrated using the Seurat pipeline selecting 3000 integration features. Counts were log normalized and scaled, and integration was performed on the 2000 genes with the highest variability across samples, using 50 dimensions. Nonlinear dimensionality reduction was conducted using UMAP, after selecting the appropriate number of principal components by determining the number of principal components that exhibits cumulative percent greater than 90% and less than 5% variation associated with. To identify cell populations, we performed iterative Louvain modularity optimization clustering from the Seurat package. We evaluated cluster quality using multiple metrics, including Gini coefficient and Silhouette score. To characterize the cell types and states defined by each cluster, we manually reviewed the differentially expressed genes, uncovered by Seurat's FindMarkers function.

### 5q deletion analysis

To differentiate del(5q) cells from non-del(5q) cells, CopyKat[23] (v1.0.8) and CaSpER[24] (v0.2.0) were applied. CopyKat excels at uncovering large-scale aberrations by effectively identifying groups of significantly deviating copy number segments across the genome. On the other hand, CaSpER utilizes BAF information to offer precise delineation of deletion boundaries by modeling heterogeneity, incorporating prior knowledge, and accounting for dependencies. This synergy helps mitigate the risk of underestimating the deletion extent or missing it altogether, as can occur with single-method approaches. Moreover, the combined analysis could potentially account for biases associated with either method individually. CaSpER was used with a non-del(5q) MDS sample that had normal karyotype[65] as reference, whereas for CopyKat, a reference composed of a combination of three healthy samples was employed. To validate the generated results, we also analyzed a non-del(5q) MDS sample using the same reference as a negative control. The raw results from CaSpER underwent filtering by extracting large-scale events using a threshold of 0.75; by raising this threshold, we increase the needed number of amplifying or deleting events in order to support the detection of such aberration, increasing the robustness of the identified CNA events. Subsequently, results

were binarized as described in the tool's methods, classifying each arm of each chromosome's arm as amplified, neutral, or deleted. For the CopyKat results, a cell clustering was performed on the results from CopyKat based on the values of copy number alterations in 220 Kbp bins of the targeted region, obtaining a cluster composed of cells with negative values. Cells exhibiting negative copy number alterations in the chr5 q15-31 region, as determined by both methods, were classified as del(5q) cells. Conversely, cells for which no alterations were identified by either method were classified as non-del(5q) cells.

### Gene Regulatory Network analysis

For each comparison, 100 transcription factors and 1000 target genes were selected based on their variability, determined by calculating the maximum absolute deviation. To determine the optimal parameters, each analysis involved a cross-validation run of SimiC[33] (v1.0.0). The resulting Gene Regulatory Networks (GRNs) were visualized using the GRN incidence matrices provided by SimiC. Histograms for different regulons were computed using the "regulon activity score" provided by SimiC, allowing to see the regulatory activity in all the cells or by cell type. Additionally, this score was utilized to calculate the regulatory dissimilarity score for the selected cell clusters.

### Differential expression analysis

Different methodologies were employed depending on the specific contrasts being examined. When the contrast allowed for the comparison of cells from different samples within each phenotype, we utilized the Libra[25] (v1.0.0) framework in combination with edgeR-LRT[66] (v4.0.6). This approach involved generating pseudobulks for each cell type, per sample, effectively aggregating the expression profiles of cells from the same phenotype. By utilizing edgeR-LRT, statistical testing was performed to identify genes that exhibited significant differential expression while accounting for the batch effect. In cases where the contrast involved a limited number of samples in any of the phenotypes, we used Libra (v1.0.0) framework in combination with MAST[67] (v1.22.0) methodology. All the results for the different contrasts were filtered by an adjusted $p$-value lower than 0.05. Gene ontology enrichments were determined using enrichR (v3.2) based on biological process and molecular function gene sets from 2023.

### Cell to cell communication analysis

The analysis of cell-to-cell communication using Liana[41] (v0.1.7) was carried out individually for each sample, comparing the healthy samples with del(5q) MDS samples. Interactions were filtered based on their statistical significance ($p$-value < 0.05), using the $p$-value from CellChat and CellPhoneDB, and on the magnitude of the interaction (log10 of the magnitude rank >5). Gene ontology enrichments were determined using enrichR (v3.2) based on biological process and molecular function gene sets from 2023.

### Statistics and reproducibility

Statistical analyses were performed in R (v4.2.2) and SimiC[33] was run in python (v3.6.9). Analysis methods for single-cell RNA sequencing data are described in the corresponding method sections and each statistical method used is specified in the corresponding figure legend. The number of human samples or cells analyzed in each subpanel is also indicated in the figure legends. No statistical method was used to predetermine sample size, and no data were excluded from the analyses performed. When possible, investigators were blinded during the data analysis (i.e., the detection of del(5q) cells was performed without prior knowledge of the karyotype result).

### Reporting summary

Further information on research design is available in the Nature Portfolio Reporting Summary linked to this article.

## Data availability

All data needed to evaluate the conclusions in the paper are present in the manuscript and/or the Supplementary Materials. The scRNA-seq data generated in this study have been deposited in the Gene Expression Omnibus database (GEO) under the accession code GSE245452. The previous publicly available data used in this study, corresponding to scRNA-seq data of CD34[+] cells from healthy age-matched individuals, are available in GEO under the accession code GSE183328. The GRCh38 assembly of the human genome used is available at NCBI, under the accession code NCBI: GCA_000001405.27. CellPhoneDB database is stored in https://github.com/ventolab/CellPhoneDB-data. The biological process and molecular function gene set libraries used for gene ontology analyses are available in https://maayanlab.cloud/Enrichr/enrich. Source data are provided as a Source Data file. Source data are provided with this paper.

## Code availability

The code and scripts for data reproducibility are available at github.com/ML4BM-Lab/MDS_5q_2023; https://doi.org/10.5281/ZENODO.10983466[68].

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

## Acknowledgements

This work was supported by the Instituto de Salud Carlos III and co-financed by ERDF A way of making Europe (PI20/01308, PI23/00516, PI19/00726, PI22/01044, PI 20/00531), CIBERONC (CB16/12/00489), RICORS TERAV (RD21/0017/0009), ERA-PerMed JTC 2019 (MEET-AML), MICCIN/AEI/10.13039/501100011033 and [RTI2018-101708-A-I00], Gobierno de Navarra (AGATA 0011-1411-2020-000010/0011-1411-2020-000011). Fundación La Caixa (GR-NET NORMAL-HIT HR20-00871), Cancer Research UK [C355/A26819], FC AECC and AIRC under the Accelerator Award Program, TRANSCAN (AECC AC 18/000002 and ISCIII), 2021 SGR 00560 (GRC) Generalitat de Catalunya; economical support from CERCA Programme/Generalitat de Catalunya, Fundació Internacional Josep Carreras and Stiftung Carreras Foundation. AECC award from the Fundación AECC (INVES19059EZPO) (T.E). H2020 Marie S. Curie IF Action, European Commission, Grant Agreement No. 898356, a Ramon y Cajal contract RYC2021-033127-I MCIN/AEI/10.13039/501100011033 (M.H). Partially funded by a Ramon y Cajal contract RYC2019-028578-I, a Gipuzkoa Fellows grant 2022-FELL-000003-01, and grant PID2021-126718OA-I00 funded by MCIN/AEI/10.13039/501100011033 (I.O.) Supported by PhD fellowships from Gobierno de Navarra (0011-0537-2019-000001) (N.B.), and (0011-0537-2020-000022) (A.D.-M.); a PhD fellowship from Ministerio de Ciencia, Innovación y Universidades (FPU18/05488) (M.A.). Funded by Sara Borrell grant (CD22/00027) from the Instituto Carlos III and NextGenerationEU (A.U-A) and personal técnico de Apoyo fellowship from the Ministry of Science and Innovation (PTA2021-020262) (P.A.-R). We particularly acknowledge the patients and healthy donors for their participation in this study, and the Biobank of the University of Navarra for its collaboration.

## Author contributions

G.S., N.B., A.D.-M., F.P., T.E., and M.H. conceived and designed the research studies; G.S., A.D.-M. and C.R.-R. performed in silico analysis of transcriptomic data; G.S., N.B., and A.D.-M. performed statistical analysis; G.S., N.B., A.D.-M., P.G.O. S.H.-D., A.A.-P., A.U.-A., B.A., C.R.-R., and M.A. analyzed and interpreted data; A.V.-Z., P.S.-M., P.A.-R., J.L.-E., A.A.-P., P.A., O.C., T.J., A.M., J.M., M.D.-C., D.V., and F.S. provided clinical samples and data; A.V.Z., P.S.-M., P.A.-R., A.U.-A., B.A., M.A. provided technical assistance; S.H.-D., and A.A.-P. provided clinical advice; T.E., I.O., M.H., and F.P. were responsible for research supervision, coordination, and strategy. N.B., A.D.-M., G.S., P.G.-O., F.P., T.E., and M.H. wrote the manuscript. All authors reviewed and approved the final version of the manuscript.

## Competing interests

The authors declare no competing interests.

## Additional information

Guillermo Serrano ⓘ [1,2,12], Nerea Berastegui ⓘ [3,4,12], Aintzane Díaz-Mazkiaran [1,3,4,12], Paula García-Olloqui ⓘ [3,4], Carmen Rodriguez-Res[1], Sofia Huerga-Dominguez[5], Marina Ainciburu ⓘ [3,4], Amaia Vilas-Zornoza[3,4], Patxi San Martin-Uriz ⓘ [3], Paula Aguirre-Ruiz[3], Asier Ullate-Agote[3], Beñat Ariceta ⓘ [3,4], Jose-Maria Lamo-Espinosa[6], Pamela Acha[7,8], Oriol Calvete[7], Tamara Jimenez ⓘ [4,9], Antonieta Molero ⓘ [8], Maria Julia Montoro[8], Maria Díez-Campelo ⓘ [4,9], David Valcarcel[8], Francisco Solé ⓘ [7], Ana Alfonso-Pierola ⓘ [4,5], Idoia Ochoa ⓘ [10,11], Felipe Prósper ⓘ [3,4,5,13] ✉, Teresa Ezponda ⓘ [3,4,13] ✉ & Mikel Hernaez ⓘ [1,4,10,13] ✉

[1]Computational Biology Program CIMA-Universidad de Navarra, Cancer Center Clínica Universidad de Navarra (CCUN), IdISNA, Pamplona, Spain. [2]Biological and Environmental Science and Engineering Division, King Abdullah University of Science and Technology (KAUST), Thuwal, Saudi Arabia. [3]Hematology-

Oncology Program, CIMA, Cancer Center Clínica Universidad de Navarra (CCUN), IdiSNA, Pamplona, Spain. [4]Centro de Investigación Biomédica en Red de Cáncer, CIBERONC, Madrid, Spain. [5]Hematology and Cell Therapy Service, Cancer Center Clínica Universidad de Navarra (CCUN), IdISNA, Pamplona, Spain. [6]Department of Orthopedics, Clínica Universidad de Navarra, Pamplona, Spain. [7]MDS Research Group, Josep Carreras Leukaemia Research Institut, Universitat Autònoma de Barcelona, Barcelona, Spain. [8]Service of Hematology, Hospital Universitari Vall d'Hebron, Barcelona; Vall d'Hebron Instituto de Oncología (VHIO), Barcelona, Spain. [9]Department of Hematology, Hospital Universitario de Salamanca-IBSAL, Salamanca, Spain. [10]Instituto de Ciencia de los Datos e Inteligencia Artificial (DATAI), University of Navarra, Pamplona, Spain. [11]Department of Electrical and Electronics engineering, School of Engineering (Tecnun), University of Navarra, Donostia, Spain. [12]These authors contributed equally: Guillermo Serrano, Nerea Berastegui, Aintzane Díaz-Mazkiaran. [13]These authors jointly supervised this work: Felipe Prósper, Teresa Ezponda, Mikel Hernaez. ✉e-mail: fprosper@unav.es; tezponda@unav.es; mhernaez@unav.es

