## [Peer Review File · Nature Communications]

Single-cell transcriptional profile of CD34+ hematopoietic progenitor cells from del(5q) Myelodysplastic Syndromes and impact of lenalidomideREVIEWER COMMENTS

Reviewer #1 (Remarks to the Author):

The manuscript by Serrano et al. offers a detailed single-cell RNA sequencing analysis of CD34+ cells from patients with 5q- syndrome. The authors performed 10X Genomics scRNA-seq on diagnostic bone marrow samples from four patients with 5q- syndrome. For comparison, they analyzed bone marrow from three age-matched healthy individuals and three patients post-lenalidomide treatment. The identification of 5q- and non-5q- cells was executed using established Copy Number Abnormality (CNA) pipelines designed for scRNA-seq data.

Unexpectedly, the transcriptional profiles of del5q cells and non-del5q cells were broadly similar, with only a limited number of genes showing significant differential expression. The authors assert that lenalidomide impacts transcriptional regulators of del5q and non-del5q cells both.

The manuscript is well-composed, and the figures are presented with clarity. While the study is predominantly descriptive, the discoveries presented are nonetheless interesting.

However, I have reservations regarding the conclusion that del5q and non-del5q cells exhibit similar transcriptional profiles and are both implicated in disease pathogenesis and therapeutic response to lenalidomide. My primary concern pertains to the classification of del5q and non-del5q cells, which relies exclusively on CNA data derived from scRNA-seq. The authors employed two analytical tools, CopyKat and CaSpER, to designate cells as del5q if identified by both. This approach prompts the question: How many supposed non-del5q cells are actually del5q cells that the pipelines failed to detect, and vice versa? Given that CNV detection from scRNA-seq lacks high sensitivity, reliance on this method alone raises doubts about the validity of the analysis. Potential misclassification of cells could explain the lack of significant transcriptional divergence. While authors compared the proportion of del5q cells between cytogenetics and the current method, this is far from validation. And the authors do not address this potential misclassification in their discussion. In my view, this issue is critical and, unless the authenticity of the del5q and non-del5q cell populations is established, I find it challenging to fully endorse the authors' conclusions.

Reviewer #2 (Remarks to the Author):

This is an interesting manuscript which evaluates the transcriptome linked to genomic deletions in chromosome 5q in the 5q del subset of patients with myelodysplastic syndromes. This is an interesting study as the authors are able to utilize single cell RNA-seq to simultaneously capture gene expression as well as chromosomal deletion in 5q simultaneously in single cells and therefore study the impact of this chromosomal change in gene expression in primary human cells. Additionally, the authors are able to study the impact of lenalidomide on the gene expression and 5q del clones using the same techniques. With that being said, the manuscript should be revised to provide some cautionary notes as follows:

-It seems that one potential major limitation of this study is that the authors are only able to discern differential expression of the mRNAs encoded by 5 genes in chromosome 5q deleted region. This may be limiting sensitivity to detect the transcriptional impact of this chromosomal alteration as well as could affect the interpretation of the samples on lenalidomide treatment. For example, the main 5q gene thought to contribute to lenalidomide sensitivity (CSNK1A) is not even detected as differentially expressed here. The authors should therefore temper the claims that 5q del and non-del(5q) cells have a similar transcriptional output.

-While the authors are able to capture 5q deletion, they do not (and likely are not able to) capture additional genetic alterations (noted in Supp. Table 1) that these patients possess. These additional genetic alterations may be present in both 5q del and non-del(5q) cells which could confound the results.

-The claims that del5q cells that persist upon lenalidomide treatment have "higher ubiquitination", signaling, etc should be revised. These claims are made based on gene expression data which are not necessarily a readout of biochemical changes in ubiquitination or signaling (and gene

expression could even be anti-correlated with some of these signaling and/or ubiquitin/degradation parameters).

RESPONSE TO REVIEWERS

Please, find below a detailed point-by-point answer to all the comments raised in the review process and an explanation on how we have addressed each of the criticisms and suggestions in the new version of the manuscript and supplemental material. Our responses are written in **blue font**, while changes in the manuscript are indicated in **red font**.

Reviewer #1 (Remarks to the Author):

The manuscript by Serrano et al. offers a detailed single-cell RNA sequencing analysis of CD34+ cells from patients with 5q- syndrome. The authors performed 10X Genomics scRNA-seq on diagnostic bone marrow samples from four patients with 5q- syndrome. For comparison, they analyzed bone marrow from three age-matched healthy individuals and three patients post-lenalidomide treatment. The identification of 5q- and non-5q- cells was executed using established Copy Number Abnormality (CNA) pipelines designed for scRNA-seq data.

Unexpectedly, the transcriptional profiles of del5q cells and non-del5q cells were broadly similar, with only a limited number of genes showing significant differential expression. The authors assert that lenalidomide impacts transcriptional regulators of del5q and non-del5q cells both.

The manuscript is well-composed, and the figures are presented with clarity. While the study is predominantly descriptive, the discoveries presented are nonetheless interesting.

However, I have reservations regarding the conclusion that del5q and non-del5q cells exhibit similar transcriptional profiles and are both implicated in disease pathogenesis and therapeutic response to lenalidomide. My primary concern pertains to the classification of del5q and non-del5q cells, which relies exclusively on CNA data derived from scRNA-seq. The authors employed two analytical tools, CopyKat and CaSpER, to designate cells as del5q if identified by both. This approach prompts the question: How many supposed non-del5q cells are actually del5q cells that the pipelines failed to detect, and vice versa? Given that CNV detection from scRNA-seq lacks high sensitivity, reliance on this method alone raises doubts about the validity of the analysis. Potential misclassification of cells could explain the lack of significant transcriptional divergence. While authors compared the proportion of del5q cells between cytogenetics and the current method, this is far from validation. And the authors do not address this potential misclassification in their discussion. In my view, this issue is critical and, unless the authenticity of the del5q and non-del5q cell populations is established, I find it challenging to fully endorse the authors' conclusions.

We thank the reviewer for the comment, and agree that the correct classification of cells into del(5q) or non-del(5q) categories is the basis of the paper and needs to be clearly explained and proven.

First of all, we would like to point out that although the CNA detection methods applied to scRNA-seq might lack sensitivity for detecting small alterations on very specific loci,

the selection criteria that we carried out to classify cells into del(5q) and non-del(5q) categories rely on the consistent detection of large consecutive events along the length of the full chromosome, which largely increases the sensitivity of the methods. Moreover, the two CNA inference algorithms used to detect del(5q) cells are complementary, in the sense that they are based on very different strategies: CopyKat relies on gene expression [1], while CaSpER relies on detection of allele frequencies [2]. As shown in Figure 2C of the revised manuscript, despite being based on different approaches, they render very similar results, with a high overlap in cells being detected as del(5q).

Nevertheless, with the objective of confirming the accuracy of our classification, we have further validated the classification through three orthogonal approaches:

1) Gene expression

Pseudobulk-driven differential gene expression (DGE) has posed as a reliable method to uncover transcriptional differences across conditions in single-cell data, as directly computing differential gene expression over the individual cells has shown to yield high false positive rates [3]. As such, when delving into the specific transcriptional differences between del(5q) and non-del(5q) cells, we opted for using a pseudobulk approach, and showed that there are few, albeit important (and statistically robust) transcriptional differences between del(5q) and non-del(5q) cell populations.

However, computing DGE over individual cells can provide “big picture” transcriptional trends and behaviors across groups of cells. Hence, to assess the correct classification of the del(5q) cells, and following the reviewer’s suggestion, we performed a DGE analysis between the individual del(5q) and non-del(5q) cells, showing that genes from chromosome 5q.13 to 5q.33 [the commonly deleted region (CDR) in our patients] were strongly enriched in those downregulated in del(5q) cells (hypergeometric test, p-value <0.05. **Figure 1A**). Similar results were obtained when the analyses were performed in individual patients (**Figure 1B**).

To account for the potential false positive rates of the above analysis, we further validated the results via permutation tests, where we performed the same DGE analysis over randomly classified cells and we observed that as the percentage of randomly assigned labels increased, the number of DE genes identified in the analysis decreased. When all labels were randomly assigned, no DE genes were detected (**Figure 1C**).

Altogether, these analyses indicate that the del(5q) and non-del(5q) labels used in the manuscript (as provided by both CopyKat and CaSpER) are strongly related to a deletion of the 5q region. Note again, that while this analysis provided further validation that identified del(5q) cells indeed harbor the 5q deletion, in the main manuscript prioritized statistical robustness and used pseudobulk-driven DGE approaches to study individual differentially expressed genes.

Figure 1. Differential expression (DE) analysis between cells labeled as del(5q) and non-del(5q) applying the Wilcox test. Volcano plot of statistical significance ($-\log_{10}(p \text{ value})$) against fold-change (\log_2 Fold-change) of gene expression between del(5q) and non-del(5q) cells. Red points represent genes with $|FC| > 0$ and adjusted p-value < 0.05 , and blue points depict genes with $|FC| < 0$ and adjusted p-value < 0.05 . A) Genes located in the CDR are significantly enriched in the set of downregulated genes in del(5q) cells (hypergeometric test, p-value < 0.05). B) Same results are observed when applying the Wilcox test per sample. D) The increase of arbitrarily shuffled cells hampers the detection of differentially expressed genes. Ultimately, when all cell labels are shuffled randomly, the analysis yields no differentially expressed genes.

We have revised the text in order to better explain the classification methods applied for del(5q) and non-del(5q) cells, and its validation based on gene expression. In particular,

we have generated an extra section called “*Identification of CD34⁺ cells harboring del(5q) in MDS patients*” that can be found in line 128 of the revised manuscript.

“Identification of CD34⁺ cells harboring del(5q) in MDS patients

Identifying single-arm copy number variations (CNVs) at the single-cell level presents challenges due to potential compensatory mechanisms of alleles, as well as to the sparse and noisy nature of single-cell data. In this study, we employed two different and complementary approaches: CopyKat, which relies on gene expression, and CaSpER, which relies on allele frequencies (see Methods). This combined strategy aimed to enhance the sensitivity and accuracy of identifying cells harboring 5q deletion. To avoid false positive detection, we only classified the cells as harboring the del(5q) if the same cell has been characterized as such by the two different algorithms. To validate this classification, we analyzed the expression pattern of genes encoded in the deleted region in individual cells. Due to the sparsity of scRNA-seq data, we were only able to detect as highly variable genes the expression of six genes, CD74, RPS14, BTF3, COX7C, HINT1 and RPS23 which were decreased in del(5q) when compared to non-del(5q) cells at sample level (Fig. 2D), further confirming our del(5q) cell classification. Once the classification was performed, we applied a wilcox test between cells classified as del(5q) and non-del(5q), revealing in the underexpressed fraction of the genes an enrichment for the genes located on the deleted locus (Fig. S2A-B). To further validate the classification, we randomly shuffled the labels from the classified cells, and repeated the same differential expression analysis, revealing how the genes located on the deleted region started to fade away (Fig. S-2C). Based on this classification, interestingly, for each individual patient, the proportion of del(5q) in the CD34⁺ progenitor cells was consistent with that obtained by karyotype in total bone marrow (Fig. 2E). “

To illustrate the robustness for the classification of the del(5q) cells, the previous Figure 1, has been added as an additional supplementary figure, being named “Supplementary Figure 2”.

Additionally, the methods regarding the classification of cells have been further detailed:

In line 507: “*To differentiate del(5q) cells from non-del(5q) cells, CopyKat²³ and CaSpER²⁴ were applied. CaSpER was used with a non-del(5q) MDS sample that had normal karyotype⁶⁰ as reference, whereas for CopyKat, a reference composed of a combination of three healthy samples was employed. To validate the generated results, we also analyzed a non-del(5q) MDS sample using the same reference as a negative control. The raw results from CaSpER underwent filtering by extracting large-scale events using a threshold of 0.75. Subsequently, results were binarized as described in the tool's methods, classifying each arm of each chromosome's arm as amplified, neutral, or deleted. For the CopyKat results, a cell clustering was performed on the results from CopyKat based on the values of copy number alterations in 220 Kbp bins of the targeted region, obtaining a cluster composed of cells with negative values. Cells exhibiting negative copy number alterations in the chr5 q15-31 region, as determined by both methods, were classified as del(5q) cells. Conversely, cells for which no alterations were identified by either method were classified as non-del(5q) cells.”*

Has been changed by:

“To differentiate del(5q) cells from non-del(5q) cells, CopyKat²³ and CaSpER²⁴ were applied. CopyKat excels at uncovering large-scale aberrations by effectively identifying groups of significantly deviating copy number segments across the genome. On the other hand, CaSpER utilizes BAF information to offer precise delineation of deletion boundaries by modeling heterogeneity, incorporating prior knowledge, and accounting for dependencies. This synergy helps mitigate the risk of underestimating the deletion extent or missing it altogether, as can occur with single-method approaches. Moreover, the combined analysis could potentially account for biases associated with either method individually. CaSpER was used with a non-del(5q) MDS sample that had normal karyotype⁶⁰ as reference, whereas for CopyKat, a reference composed of a combination of three healthy samples was employed. To validate the generated results, we also analyzed a non-del(5q) MDS sample using the same reference as a negative control. The raw results from CaSpER underwent filtering by extracting large-scale events using a threshold of 0.75; by raising this threshold, we increase the needed number of amplifying or deleting events in order to support the detection of such aberration, increasing the robustness of the identified CNA events. Subsequently, results were binarized as described in the tool's methods, classifying each arm of each chromosome's arm as amplified, neutral, or deleted. For the CopyKat results, a cell clustering was performed on the results from CopyKat based on the values of copy number alterations in 220 Kbp bins of the targeted region, obtaining a cluster composed of cells with negative values. Cells exhibiting negative copy number alterations in the chr5 q15-31 region, as determined by both methods, were classified as del(5q) cells. Conversely, cells for which no alterations were identified by either method were classified as non-del(5q) cells.”

2) Single nucleotide polymorphisms detection

To further prove that cells labeled as del(5q) indeed harbored the deletion, we aim at uncovering potential loss of heterozygosity (LOH) of the 5q region of such cells. Thus, we performed germline variant detection using VarScan on chromosome 5 for each patient. We considered polymorphisms with a minimum of 8 reads in non-del(5q) cells, then depicted their frequencies in both non-del(5q) and del(5q) cells. Our hypothesis was grounded in the assumption that accurate classification should result in SNP frequencies within the deleted region approaching either 0% or 100% in del(5q) cells, while SNP frequencies in non-del(5q) cells were expected to distribute around 0%, 50% or 100%.

Our analyses reveal a statistically significant reduction of SNPs with 50% frequency in del(5q) cells compared to non-del(5q) cells within the deleted region, thereby further validating the accuracy of our classification (**Figure 2**).

Figure 2. Frequency of SNPs located in chromosome 5 detected by VarScan per sample. Y axis depicts SNP frequency, X axis shows positions of chromosome 5 and the zone depicted in blue corresponds to the deleted region identified by karyotype. Del(5q) cells have significantly less SNPs with around 50% of frequency compared to non-del(5q) cells (Kolmogorov-Smirnov test, p -value < 0.05). SNPs present in more than 8 reads have been considered in the analysis.

3) Complementary CNV inference methods

To further validate our classification, we employed a third CNV inference algorithm called inferCNV [4]. Unlike CopyKat and CaSpER, inferCNV utilizes a distinct methodological approach, examining gene expression intensity across positions of the tumor genome compared to ‘normal’ reference cells, to provide a score representing the likelihood of each cell harboring the deletion. We thus applied inferCNV to each patient using a non-del(5q) MDS patient as a control, obtaining scores representing the sum of gene expression within the 5q13-5q33 region for each cell. Thus, lower scores indicated a higher likelihood of being a del(5q) cell.

Figure 3A depicts the empirical cumulative distribution function (ecdf) for del(5q) and non-del(5q) cells classified by CopyKat and CaSpER. Non-del(5q) cells have statistically significant higher inferCNV scores, whereas del(5q) cells exhibited a strong bias towards lower scores, suggesting a high likelihood of the latter being del(5q) cells (K-S test, p -value $< 2.2e-16$ for all patients). Additionally, as observed in **Figure 3B**, del(5q) cells

were significantly enriched in those cells that inferCNV predicted to have a higher del(5q) likelihood (gsea test, p-value $<2.2e-16$).

Figure 3. Del(5q) cells identified by CopyKat and CaSpER are enriched in cells with higher 5q likelihood inferred by inferCNV. A) Differential empirical cumulative distribution function (ecdf) of del(5q) and non-del(5q) cells classified by CopyKat and CaSpER, where distribution of del(5q) cells is skewed towards lower inferCNV scores (Kolmogorov-Smirnov test, p-value $<2.2e-16$). B) The GSEA approach reveals an enrichment of del(5q) cells classified by CopyKat and CaSpER among the top-ranked cells, ordered from lower to higher inferCNV scores.

Overall, this analysis, along with the gene expression and SNP detection analyses, further reinforces the classification of del(5q) and non-del(5q) cells performed by CopyKat and CaSpER.

Furthermore, in response to the reviewer's suggestion, we have incorporated a new paragraph into the discussion section to address the potential misclassification of del(5q) and non-del(5q) cells.

- In line 369: *“Using scRNAseq data from CD34+ cells from patients with del(5q) MDS, we have been able to identify cells with del(5q) and non-del(5q), which enabled us to compare the transcriptional profile and GRNs of both populations within the same patient, thereby mitigating potential confounding factors associated with interindividual variability and microenvironmental influences, and thus made strides in the understanding of the molecular pathogenesis of del(5q) MDS. Single cell RNA-seq coupled with CNA detection algorithms offers the opportunity to link the genetic information to the transcriptional profile of each individual cell. Nonetheless, this type of analyses remain challenging due to the sparse and noisy nature of single-cell data⁵¹. To ensure robust findings we used two CNA detection methods, one based on transcript abundance²³ and another utilizing allelic imbalance in heterozygous SNPs²⁴. While expression-based approaches operate under the assumption that amplifications or deletions generally result in up- or down-regulation of genes within the affected region of the genome, allele-based methodologies focus on analyzing deviations in heterozygous allele frequency. We acknowledge the potential for minor misclassification due to the inherent complexity of applying CNA detection algorithms to scRNA-seq data, however, we believe that this would be minimal, as clearly demonstrated in the results section (supplementary figure 2). Our confidence in employing two complementary methods stems from the belief that this approach enhances classification accuracy and mitigates false-positive results arising from local expression variations unrelated to genomic copy numbers, as well as from data sparsity and allele-specific transcriptional stochasticity^{23,24}”.*

Reviewer #2 (Remarks to the Author):

This is an interesting manuscript which evaluates the transcriptome linked to genomic deletions in chromosome 5q in the 5q del subset of patients with myelodysplastic syndromes. This is an interesting study as the authors are able to utilize single cell RNA-seq to simultaneously capture gene expression as well as chromosomal deletion in 5q simultaneously in single cells and therefore study the impact of this chromosomal change in gene expression in primary human cells. Additionally, the authors are able to study the impact of lenalidomide on the gene expression and 5q del clones using the same techniques.

We sincerely acknowledge the reviewer's comments and the thorough review of our manuscript, which we believe provides a valuable perspective on important aspects that require further consideration.

With that being said, the manuscript should be revised to provide some cautionary notes as follows:

-It seems that one potential major limitation of this study is that the authors are only able to discern differential expression of the mRNAs encoded by 5 genes in chromosome 5q deleted region. This may be limiting sensitivity to detect the transcriptional impact of this chromosomal alteration as well as could affect the interpretation of the samples on lenalidomide treatment. For example, the main 5q gene thought to contribute to lenalidomide sensitivity (CSNK1A) is not even detected as differentially expressed here. The authors should therefore temper the claims that 5q del and non-del(5q) cells have a similar transcriptional output.

We understand and appreciate the concern regarding the limited sensitivity to detect differentially expressed genes in the commonly deleted region (CDR) of chromosome 5. We recognize that this limitation could impact our ability to fully capture the transcriptional impact of the chromosomal alteration and, consequently, could affect the interpretation of the effects of lenalidomide treatment. The specific point about the non-detection of CSNK1A as a differentially expressed gene is particularly relevant, and we appreciate highlighting this issue.

As previously described in our response to Reviewer #1, we conducted additional analyses to explore this aspect further. Specifically, we performed differential expression analysis at the level of individual cells (via the Wilcox test and Seurat [5], which revealed a greater number of differentially expressed genes between del(5q) and non-del(5q) cells. Note however, that this analysis generally yields a high false positive rate, and hence, in the manuscript we erred on the side of statistical robustness and performed pseudobulk-driven differential gene expression, at the cost of potentially missing some genes, such as CSNK1A. Indeed, among the genes that were downregulated in del(5q) cells via Wilcox test, we identified an enrichment of genes located within the CDR of chromosome 5 (**Figure 1A-B**). Among them, CSNK1A, which is known to contribute to lenalidomide sensitivity, was found to be significantly underexpressed in del(5q) cells compared to both non-del(5q) cells and healthy donor cells (**Figure 4A**, Wilcox test, p-value <0.05). This observation suggests that the haploinsufficiency of CSNK1A, which confers increased sensitivity to lenalidomide, is indeed reflected in the underexpression of this gene in del(5q) cells.

Figure 4. Differential expression analysis of CSNK1A using the Wilcoxon test. A) CSNK1A was significantly downregulated in del(5q) cells in respect to non-del(5q) and healthy cells (Wilcoxon test, p-value <0.05). B) CSNK1A was significantly underexpressed in del(5q) cells in all patients (Wilcoxon test, p-value <0.05).

-While the authors are able to capture 5q deletion, they do not (and likely are not able to) capture additional genetic alterations (noted in Supp. Table 1) that these patients possess. These additional genetic alterations may be present in both 5q del and non-del(5q) cells which could confound the results.

We appreciate the reviewer's insightful comment, acknowledging that the additional genetic alterations observed in our patients might be present in both del(5q) and non-del(5q) cells, potentially having an effect in the transcriptional profile of the cells.

However, it is important to note that only three subtypes of MDS are classified by genetic abnormalities, as outlined by the WHO 2022 classification: 1) MDS with low blasts and isolated 5q deletion (MDS-5q), 2) MDS with low blasts and SF3B1 mutation (MDS-SF3B1), and 3) MDS with biallelic TP53 inactivation (MDS-biTP53). Interestingly, around 70% of del(5q) MDS patients exhibit additional somatic mutations in addition to the deletion, while approximately 30% do not display such mutation [6]. Nevertheless, despite the variability in these additional somatic mutations, del(5q) patients form a clinically homogeneous subgroup. Hence, the primary objective of our study was to elucidate the transcriptional impact of the deletion, with the final goal of unraveling the molecular pathogenesis of del(5q) MDS, regardless of additional alterations.

In MDS patients, del(5q) and non-del(5q) cells are mixed without clear distinguishing cell surface markers that are feasible for fluorescence-activated cell sorting (FACS) enrichment. To overcome this challenge, and in order to identify specific cells harboring the deletion, it is imperative to utilize single-cell methods coupled with CNV detection algorithms. While droplet-based sequencing allows for the simultaneous transcriptomic profiling of thousands of cells, current methodologies inherently provide sequence information limited to the 3' end of the transcript, which restricts the ability to jointly genotype somatic mutations. To address this limitation, Genotyping of Transcriptomes (GoT) was developed [7] to link genotypes of expressed genes to transcriptional profiling of thousands of single cells. However, a challenge arises from the dependency of this genotyping approach on captured mRNA transcripts, which is inherently limited by the expression level of the targeted gene. As a proof of concept and in light of the observed mutational diversity within our patient cohort, as well as the low expression level of the mutated genes (*TET2*, *DNMT3A*, *ASXL1* and *SF3B1*), we chose to try to genotype a single mutation in an individual patient.

We performed GoT in CD34+ bone marrow cells of Patient_3 (SMD35303) with *TET2* mutation (NM_001127208.2:exon11:c.5951dupT:p.V1984fs). We obtained genotyping information of 28% of cells, even for this very lowly expressed gene (**Figure 5A**). Although we detected *TET2* mutated cells in all hematopoietic progenitors (**Figure 5B**), and in both del(5q) and non-del(5q) cells (**Figure 5C-D**), these were positively enriched in cells harboring the deletion (hypergeometric test, p-value<0.05).

Figure 5. Distribution of *TET2*-mutated cells within del(5q) and non-del(5q) CD34+ progenitors from Patient_3 (SMD35303) after performing GoT. A) Normalized expression of *TET2* gene in Patient_3. B) UMAP depicting the distribution of *TET2*-mutated and WT cells in CD34+ hematopoietic progenitors. C) Number and (D) proportion of *TET2*-mutated and WT cells in del(5q) and non-del(5q) hematopoietic progenitors.

As the reviewer notes, the mutation is present in both del(5q) and non-del(5q) cells, and thus, could contribute to transcriptional similarity observed between both cell populations. However, the significant diversity of possible additional mutations and the biologically relatively homogeneous group formed by del(5q) patients might indicate the presence of other additional factors, such as a disrupted microenvironment or additional unknown genetic lesions that play a key role in the observed similarities. Future studies with larger cohorts of cases may elucidate the underlying nature of these observed transcriptional similarities.

Moreover, we have included a new paragraph in the discussion section, discussing the potential additional alterations that could lead to the observed transcriptional similarities in del(5q) and non-del(5q) cells.

- In line 398: “*The comparison between the transcriptional profiles of del(5q) cells and non-del(5q) within the same patient provides new insights regarding the impact of 5q deletions on pathogenesis of del(5q) MDS. The presence of similar transcriptional alterations in both del(5q) and non-del(5q) suggests that in del(5q) MDS patients, both types of cells are implicated in the disease and contribute to the promotion of aberrant hematopoietic differentiation. Nevertheless, the boosted aberrant activity of some regulons in del(5q) cells, may*

exert a more prominent role of these cells in the aberrant hematopoiesis of these patients. Furthermore, this fact might indicate the presence of other additional factors, such as a disrupted microenvironment, as evidenced by recent studies elucidating its impact on the initiation and progression of MDS^{53,54}. Moreover, the presence of additional shared genetic lesions between del(5q) and non-del(5q) cells may play a key role in the observed transcriptional similarities. Future studies with larger cohorts of cases may elucidate the underlying nature of these observed transcriptional similarities.”

-The claims that del5q cells that persist upon lenalidomide treatment have “higher ubiquitination”, signaling, etc should be revised. These claims are made based on gene expression data which are not necessarily a readout of biochemical changes in ubiquitination or signaling (and gene expression could even be anti-correlated with some of these signaling and/or ubiquitin/degradation parameters).

We thank the reviewer for this relevant comment. It is true that our results are solely based on gene expression and that measuring protein abundance provides information that is not apparent from gene expression, and that is crucial for the description of the state of a biological system. However, it is noteworthy that many studies, such as those highlighted in the following high-impact works [8,9], often utilize measured mRNA concentrations to linearly approximate corresponding protein levels. This approach relies on the fact that mRNA levels (unlike protein abundances) are relatively easy to determine, which enables precise and high-throughput measurements.

In MDS patients, del(5q) and non-del(5q) cells are admixed without clear distinguishing cell surface markers amenable to fluorescence-activated cell sorting (FACS) enrichment. To overcome the challenge of resolving admixed del(5q) and non-del(5q) cells in primary human samples, it becomes imperative to perform single-cell multi-omics methods followed by CNA analysis to effectively link genotypes to transcriptional profiles. Although multiple pioneering approaches have started to profile small numbers of intracellular and intranuclear proteins along with the transcriptional profile at single-cell resolution [9], to date, these approaches are not compatible with the low number of CD34+ cells obtained from primary del(5q) MDS samples. We would like to emphasize the challenge inherent in obtaining this type of primary samples compounded by the low number of CD34+ progenitors in the bone marrow.

Nevertheless, we agree with the reviewer and thus, we have revised the text and replaced some of the original sentences, limiting our claims to gene expression observations in order to avoid overstatements:

- In line 172: *“Genes overexpressed in del(5q) cells were enriched in cell cycle and mitosis-related signatures, such as DNA replication and mitotic nuclear division, and showed increased DNA repair, suggesting that loss of 5q confers increased proliferative potential. Additionally, del(5q) erythroid progenitors, LMPPs, GMPs, DCs, and monocyte progenitors showed a positive enrichment in p53 signaling pathway, and an enrichment in apoptosis-related pathways was detected in LMPP, MEP and late erythroid progenitors, but not in early erythroid progenitor cells. Our results are in line with the increased levels of apoptosis described for del(5q) patients^{28,29,30}. Downregulated genes showed enrichment in*

processes related to ribosomes and translation in all hematopoietic progenitors, in line with previous works that have described del(5q) MDS as a ribosomopathy”

has been replaced by:

“Genes overexpressed in del(5q) cells were enriched in cell cycle and mitosis-related signatures, such as DNA replication and mitotic nuclear division, and showed, increased expression of DNA repair related genes, suggesting that loss of 5q confers increased proliferative potential. Additionally, del(5q) erythroid progenitors, LMPPs, GMPs, DCs, and monocyte progenitors showed significant upregulation of genes involved in the p53 signaling and, genes involved in the apoptosis pathway were significantly upregulated in was detected in LMPP, MEP and late erythroid progenitors, but not in early erythroid progenitor cells. Our results are in line with the increased levels of apoptosis described for del(5q) patients^{28,29,30}. Downregulated genes showed enrichment in processes related to in ribosomes and translation related pathways in all hematopoietic progenitors, in line with previous works that have described del(5q) MDS as a ribosomopathy”

- In line 292: *“Furthermore, hematopoietic progenitors exhibited enhanced erythropoietin signaling and an increased expression of genes involved in erythroid differentiation after treatment”* has been replaced by: *“Furthermore, hematopoietic progenitors exhibited an increased expression of genes involved in erythroid differentiation and erythropoietin signaling after treatment”*.
- In line 309: *“Continue exhibiting ribosome-related alterations”* has been replaced by: *“continue exhibiting altered expression of ribosome-related genes”*.
- In line 323: *“Generally exhibited higher ubiquitination, phosphatidylinositol signaling, autophagy, and apoptosis when compared to del(5q) cells at diagnosis”* has been replaced by: *“Generally exhibited significant upregulation of genes involved in ubiquitin and phosphatidylinositol signaling and, autophagy and apoptosis pathways”*.
- In line 327: *“However, these cells showed reduced ribosomal and mitochondrial translation compared to diagnosis, along with diminished DNA repair capacity”*, has been replaced by: *“However, these cells showed reduced expression of genes implicated in ribosomal and mitochondrial translation compared to diagnosis, along with diminished expression of DNA repair associated genes”*.
- In line 342: *“Moreover, these cells also exhibited an increased erythropoietin signaling as a result of lenalidomide treatment when compared to the NR cells”*, has been replaced by: *“Moreover, these cells also exhibited an increased expression of genes involved in erythropoietin signaling when compared to the NR cells”*.

References

1. Gao, R. *et al.* Delineating copy number and clonal substructure in human tumors from single-cell transcriptomes. *Nat Biotechnol* **39**, 599–608 (2021).
2. Serin Harmanci, A., Harmanci, A. O. & Zhou, X. CaSpER identifies and visualizes CNV events by integrative analysis of single-cell or bulk RNA-sequencing data. *Nat Commun* **11**, (2020).
3. Squair, J. W *et al.* Confronting false discoveries in single-cell differential expression. *Nat Commun* **12**, 5692 (2021).
4. Patel, A. P. *et al.* Single-cell RNA-seq highlights intratumoral heterogeneity in primary glioblastoma. *Science* **344**, 1396–1401 (2014).
5. Hao, Y. *et al.* Dictionary learning for integrative, multimodal and scalable single-cell analysis. *Nat Biotechnol.* **42**, 293-304 (2023).
6. Bruzzese, A. *et al.* Myelodysplastic syndromes del (5q): Pathogenesis and its therapeutic implications. *European Journal of Haematology* (2024).
7. Nam, A. S. *et al.* Somatic mutations and cell identity linked by Genotyping of Transcriptomes. *Nature* **571**, 355-360 (2019).
8. Colla, S. *et al.* Telomere dysfunction drives aberrant hematopoietic differentiation and myelodysplastic syndrome. *Cancel Cell* **27**, 644-657 (2015).
9. Ganan-Gomez, I. *et al.* Stem cell architecture drives myelodysplastic syndrome progression and predicts response to venetoclax-based therapy. *Nat Med* **28**, 557-567 (2022).
10. Blair, J. D. *et al.* Phospho-seq: Integrated, multi-modal profiling of intracellular protein dynamics in single cell. *Biorxiv* (2023).

REVIEWERS' COMMENTS

Reviewer #1 (Remarks to the Author):

The authors did an excellent job in addressing my concern and I think the manuscript significantly improved.

Reviewer #2 (Remarks to the Author):

The authors have addressed my initial concerns and questions.